# Linear SCM Identification in the Presence of Confounders and Gaussian Noise

**Vahideh Sanjaroon**
Department of Electrical & Computer Engineering
Isfahan University of Technology
Isfahan, 8415683111 Iran
`vahidehsanjaroon@gmail.com`

**Pouria Ramazi**
Department of Mathematics & Statistics
Brock University
St. Catharines, ON L2S 3A1, Canada
`pramazi@brocku.ca`

## Abstract

Noisy linear structural causal models (SCMs) in the presence of confounding variables are known to be identifiable if all confounding and noise variables are non-Gaussian and unidentifiable if all are Gaussian. The identifiability when only some are Gaussian remains unclear. We show that, in the presence of Gaussian noise, a linear SCM is uniquely identifiable provided that *(i)* the number of confounders is at most the number of the observed variables, *(ii)* the confounders do not have a Gaussian component, and *(iii)* the causal structure of the SCM is known. If the third condition is relaxed, the SCM becomes finitely identifiable, belonging to a set of at most $n!$ linear SCMs, where $n$ is the number of observed variables. The confounders in all of these $n!$ SCMs share the same joint probability distribution function (PDF), which we obtain analytically. For the case where both the noise and confounders are Gaussian, we provide further insight into the existing counter-example-based unidentifiability result and demonstrate that every SCM with confounders can be represented as an SCM without confounders but with the same joint PDF.

## 1 Introduction

Achieving a desired state in a system requires appropriate interventions, which necessitate a causal model rather than a purely probabilistic or correlation-based approach. This is because interventional queries are not always answered by probabilistic models. A Structural Causal Model (SCM) (Peters et al., 2017) formalizes the common sense intuition of the notion of causality. In its general form, an SCM consists of a number of random variables, and enforcing each variable to be a deterministic function of a single noise variable and a subset of the other random variables, known as its *causes*. The noise variables are assumed to be disjoint.

An SCM allows systematic answers to observation queries (Spirtes et al., 2000), such as the probability of $X$ given $Y$, intervention queries, like the probability of $X$ if $Y$ is intervened upon, and counterfactual queries, for example, what is the probability of $X$ had $Y$ taken a different value. However, first the structure of the SCM, namely, the functions assigned to each variable, must be estimated from data, a task referred to as *causal discovery* (Chen et al., 2021; Yang et al., 2022; Xie et al., 2023; Agrawal et al., 2021). The available data are usually limited to observations rather than interventions due to infeasibility or high costs of experiments. The question then is up to what extent, an SCM can be estimated from observation data. More specifically, given the joint probability distribution function (PDF) of the random variables, is it possible to uniquely obtain the functions of an SCM; namely, is the SCM *identifiable* (Hyvärinen et al., 2023)?

There is yet another intrinsic challenge in answering the identifiability question: the existence of *confounders*, i.e., unobserved variables that affect the considered observed variables in the SCM. Indeed, it is often the case that several variables have strong associations without any causal relationship, indicating the presence of one or more confounders (Spirtes et al., 2000; Witty et al., 2021). It is nearly impossible to claim that all causes of a specified process are considered; hence, the consideration of confounders is inevitable. While there has been some progress on the identifiability of SCMs in the presence of confounders in the general case where the assigned functions can be

nonlinear (Shimizu, 2014; Hyvärinen et al., 2023; Wu et al., 2022; Witty et al., 2021; Agrawal et al., 2021), this problem remains open and challenging due to the information inaccessibility regarding the confounders and the complexity of the analysis. Thanks to the linearization techniques around the operating point of a system, the assigned functions are often supposed linear, resulting in a *linear SCM*, resulting in the system of equations $\underline{X} := A\underline{X} + B\underline{H} + C\underline{Z}$, where random variables $\underline{X}$, $\underline{H}$, $\underline{Z}$ are the observed variables, confounders, and noise variable, and $A$, $B$, $C$ are the coefficient matrices (Eriksson & Koivunen, 2003). The identifiability of the SCM then reduces to determining the triple $(A, B, C)$ from the joint PDF of the observed variables $\mathbb{P}(\underline{X})$.

## 1.1 RELATED LITERATURE ON LINEAR SCMs

The base of many existing work on the identification of linear SCMs is *Independent Component Analysis (ICA)* (Hyvärinen & Oja, 2000) applied to the so-called *ICA model* where the observed variables $\underline{X}$ are assumed to be a linear combination of some independent components, denoted by the vector $\underline{S}$, and are governed by the equation $\underline{X} := A\underline{S}$, where $A$ is known as the *mixing matrix*. In fact, ICA is developed based on Darmois-Skitovich Theorem (Pavan & Miranda, 2018). It is proven that a mixing matrix $A$ for a non-Gaussian ICA model is identifiable up to the permutation and scaling of its columns. Namely, two mixing matrices $A$ and $\tilde{A} = A\Gamma_m P_m$ may result in the same observation PDF $\mathbb{P}(\underline{X})$, where $\Gamma_m$ and $P_m$ are scaling and permutation matrices. However, if two mixing matrices $A$ and $\hat{A}$ result in the same $\mathbb{P}(\underline{X})$ and one column of $A$ is independent of the columns in $\hat{A}$, then at least one of the sources are Gaussian (Taleb & Jutten, 1999). For the case where the number of sources exceeds the observable variables, the *overcomplete ICA* is developed (Podosinnikova et al., 2019).

Clearly, the idea of ICA are also applicable to the framework of SCMs. However, based on the assumptions and model limitations, researchers are suggesting new algorithms (Xie et al., 2023; Yang et al., 2022; Salehkaleybar et al., 2020; Adams et al., 2021). In the presence of unobserved variables (also called *latent* variables) the identifiability was shown in (Xie et al., 2023) under the assumption that every latent variable has at least two pure measurement variables (those that do not have an observable cause and have at most one latent cause). Yang et al. (2022) introduce two different classes of linear SCMs, i.e., P-SCM and its subset D-P-SCM, in which some latent variables are considered as noise and influence only one variable. It is proved that in the set of P-SCM, a D-P-SCM is not identifiable. Under the faithfulness assumption and using ICA, (Salehkaleybar et al., 2020) show that non-Gaussian noise and confounders, the SCM is unidentifiable but there are a finite number of SCMs that result in the PDF of the observed variables and that these SCMs can be obtained.

The authors in (Xie et al., 2022) study the identification of Hierarchical Structures, where the observed variables do not interact and are caused only by the latent variables. Adams et al. (2021) show that if strong non-redundancy and bottleneck faithfulness hold, the SCM is identifiable up to trivialities. Giraud & Tsybakov (2012) and Frot et al. (2019) study linear systems with sparsely related observed variables. Giraud & Tsybakov (2012) estimate the number of latent variables and the conditional graphical model structure among the observed variables, and Frot et al. (2019) estimate the Markov equivalence class of the directed acyclic graph over the observed variables.

Moreover, Shimizu et al. (2009); Cai et al. (2019); Xie et al. (2020); Anandkumar et al. (2013) estimate the causal structures of hidden variables with the assumption that the observable variables are not caused by each other. Nowzohour et al. (2017) study bow-free graph in which there cannot be both a directed edge and a bow (confounder) between the same pair of variables. Chen et al. (2021) consider linear non-Gaussian SCMs in the presence of latent confounders and present hybrid algorithms to determine unknown causal relations by means of regression, independence tests, Traid conditions (Cai et al., 2019), and over complete ICA algorithm. Over complete ICA has a closed form solution for the mixing matrix in specific cases using higher order cumulants (Cai et al., 2023). Xie et al. (2020) propose a Generalized Independent Noise (GIN) condition to discover the causalities under the assumption that there is no direct edge between observed variables, which is the same assumption considered in the structure in (Squires et al., 2022). D'Amour (2019) present examples in which the system is not identifiable, such as the special case of a Gaussian linear SCM. Li et al. (2024) provide a sufficient condition for the identifiability of a linear SCM with observed and latent variables under the faithful assumption. The condition requires having either for each latent variable one *pure child*, that is an observed variable that is caused only by that latent variable and itself does not cause any other variable, or no pure child but a non-Gaussian distribution for a subset of noise variables.

## 1.2 CONTRIBUTION

In this paper, we distinguish between noise, which we assume to be Gaussian and affects only one observed variable, and confounders, which can be either all Gaussian or all non-Gaussian and can affect multiple observed variables. Our contribution and distinction from existing work are as follows:

*(1)* For the case where both the noise and confounders are Gaussian, we show that there are infinitely many SCMs consistent with the same observed PDF–Theorem 1. This provides further insight into the existing counter-example-based unidentifiability results (Eriksson & Koivunen, 2004; Hoyer et al., 2008). *(2)* We then show that in order for the SCM to be finitely identifiable–that is only a finite number of SCMs result in the observed PDF–it is sufficient for the confounders to be non-Gaussian (Theorem 2), not necessarily also the noise as in (Salehkaleybar et al., 2020). The work in (Li et al., 2024) also allows some noise variables to be Gaussian, however unique identifiability results rely on restrictive conditions, such as having a *generalized pure pair*. In contrast, our work demonstrates that while identifiability may not be unique, it is possible to identify a finite (equivalence) set of causal structures without relying on extra assumptions. *(3)* We provide an upper bound on the size of this finite set in the case of Gaussian noise and non-Gaussian confounders. We provide an algorithm that given an SCM, obtains all other SCMs in its equivalence set–Algorithm 1–those who share the same observed PDF as the given SCM. *(4)* Thanks to the novel approach of using characteristic functions in this paper, we obtained precisely the PDF of the confounders–(3) in Theorem 2. *(5)* For Gaussian noise and non-Gaussian confounders, distinct SCMs within the finite set must have different causal orders. Thus, unique identifiability would be achieved, should the causal order be specified. In contrast, as illustrated in (Salehkaleybar et al., 2020), when all variables are non-Gaussian, it is possible for two SCMs to have the same causal order.

In Section II, we present the problem formulation. Section III addresses Gaussian Structural Causal Models (SCMs). In Section IV, we examine the identifiability of non-Gaussian SCMs. A comparison of identifiability results based on noise and confounder distributions is provided in Section V. Numerical examples are discussed in Section VI. Finally, Section VII concludes the paper.

## 2 SYSTEM MODEL AND PRELIMINARIES

### 2.1 NOTATIONS

Vectors are denoted by an underline, e.g., $\underline{t}$. Deterministic variables are lowercase and random variables are uppercase. Matrices are uppercase and boldface, e.g., $\boldsymbol{A}$. The notation $(\cdot)^\top$ represents the matrix transpose and $E[\cdot]$ denotes the expected value operator. A Gaussian random distribution with expected value $\boldsymbol{\mu}$ and covariance matrix $\boldsymbol{\Sigma}$ is denoted by $\mathcal{N}(\boldsymbol{\mu}, \boldsymbol{\Sigma})$. $\boldsymbol{I}_k$ is the identity matrix with dimension $k$. The component located at the $i^{\text{th}}$ row and $j^{\text{th}}$ column of matrix $\boldsymbol{M}$ is represented by $M_{ij}$. The cardinality of a set $\mathcal{X}$ is denoted by $|\mathcal{X}|$.

### 2.2 PROBLEM FORMULATION

Consider the linear SCM $\mathfrak{C}$ defined by

$$\underline{X} := \boldsymbol{A}\underline{X} + \boldsymbol{B}\underline{H} + \boldsymbol{C}\underline{Z} \tag{1}$$

where *observed vector* $\underline{X} \in \mathbb{R}^{n \times 1}$ is a vector of zero-mean observed variables $X_i$, $i = 1, \ldots, n$, with $n \geq 2$, $\underline{H} \in \mathbb{R}^{k \times 1}$, $k \geq 0$, is a vector of mutually independent zero-mean *confounding variables (or confounders)*, $\underline{Z} \in \mathbb{R}^{n \times 1}$ is a vector of zero-mean mutually independent Gaussian *noise variables* with $\underline{Z} \sim \mathcal{N}(\underline{0}_{1 \times n}, \boldsymbol{I}_n)$, $\boldsymbol{A} \in \mathbb{R}^{n \times n}$ is a matrix with zero diagonal components in which $A_{ij}A_{ji} = 0$ due to the casual structure, $\boldsymbol{B} \in \mathbb{R}^{n \times k}$, and $\boldsymbol{C} \in \mathbb{R}^{n \times n}$ is a positive definite diagonal matrix. Without loss of generality, we assume $E[\underline{H}\underline{H}^\top] = \boldsymbol{I}_k$.

The joint PDF of the observed vector $\underline{X}$ and confounder vector $\underline{H}$ are denoted by $\mathbb{P}(\underline{X})$ and $\mathbb{P}(\underline{H})$, respectively. It follows that (1) can be written as

$$\underline{X} = \boldsymbol{M}\underline{H} + \boldsymbol{Q}\underline{Z} \tag{2}$$

where $\boldsymbol{Q} = (\boldsymbol{I}_n - \boldsymbol{A})^{-1}\boldsymbol{C}$ is a positive definite matrix with $Q_{ij}Q_{ji} = 0$ for every $i \neq j$, and $\boldsymbol{M} = (\boldsymbol{I}_n - \boldsymbol{A})^{-1}\boldsymbol{B}$. The parameters in (1) can be derived according to $(\boldsymbol{M}, \boldsymbol{Q})$ as follows. $\boldsymbol{C}$ is a

diagonal matrix where its diagonal elements are equal to the diagonal elements of $Q$. Then $A$ and $B$ are computed, respectively, as $A = I_n - CQ^{-1}$ and $B = M - AM$. Thus, an SCM $\mathfrak{C}$ can be represented either by the ordered pair $\mathfrak{C}(M, Q)$ or ordered triple $\mathfrak{C}(A, B, C)$.

The SCM induces a (causal) DAG over the observed and confounders, where there is a link from observed variable $X_i$ (resp. confounder $H_i$) to observed variable $X_j$ if $A_{ji} \neq 0$ (resp. $B_{ji} \neq 0$). In this case, we say that $X_i$ (resp. $H_i$) *causes/influences* $X_j$. The confounders are not linked to each other.

**Remark 1 (Canonical model)** *The SCM defined by* (1) *matches the so-called* canonical model *in which each confounder is a root variable, i.e., with no parent. For SCMs with non-Gaussian confounders where the confounders are not root variables, an observationally and causally equivalent canonical model can be obtained using Algorithm A in (Hoyer et al., 2008) (by skipping step 3 in the algorithm).*

**Definition 1 (Informative observable variables)** *A set of informative observed variables (of size $m$) is a set of observed variables corresponding to any $m$ linearly independent rows of matrix $M$. The remaining $n - m$ observed variables are called* non-informative *(with respect to this set).*

**Definition 2 (Causal (structure) order)** *A causal order for SCM $\mathfrak{C}$ denoted by $\mathcal{O}_{\mathfrak{C}}$ is defined as the topological order of the variables in the induced DAG, that is, an ordering over the observed variables, i.e., $X_{i_1} \succ X_{i_2} \ldots \succ X_{i_n}$ where $\{X_{i_1}, \ldots, X_{i_n}\} = \{X_1, \ldots, X_n\}$, such that there is no path from $X_{i_{k+1}}$ to $X_{i_k}$ in the induced DAG for every $k \in \{1, 2, \ldots, n\}$.*

The causal structure order is not necessarily unique, because the induced DAG may not be connected. The set $\mathfrak{O}_{\mathfrak{C}}$ denotes the set of all possible causal orders for the SCM $\mathfrak{C}$.

We use the concept of equality in distribution, denoted as $\underline{X} \overset{d}{=} \underline{Y}$, for random variables $\underline{X}$ and $\underline{Y}$ that have the same distribution (Definition 12). In particular, when $\underline{X}$ and $\underline{X}'$ are the observed variables of SCMs $\mathfrak{C}$ and $\mathfrak{C}'$, $\underline{X} \overset{d}{=} \underline{X}'$ implies that they share the same support and have identical distributions, i.e., $\mathbb{P}(\underline{X}) = \mathbb{P}(\underline{X}')$.

**Definition 3 (SCM finite identifiability)** *SCM $\mathfrak{C}$ in the form of* (1) *is* finitely identifiable *if and only if there exists a finite number of SCMs $\mathfrak{C}'$ such that $\underline{X} \overset{d}{=} \underline{X}'$ where $\underline{X}$ and $\underline{X}'$ are the observed variables in SCMs $\mathfrak{C}$ and $\mathfrak{C}'$. The set of all such SCMs is called the* finite equivalence class *of SCM $\mathfrak{C}$. The SCMs in this class are said to be* equivalent *to each other.*

Unique identifiability is the same finite identifiability but when the cardinality of the equivalence class equals one. The following is the equivalent definition used in the literature.

**Definition 4 (SCM unique identifiability)** *SCM $\mathfrak{C}(A, B, C)$ in the form of* (1) *is uniquely identifiable if and only if for every other SCM $\mathfrak{C}'(A', B', C')$ it holds that*

$$\underline{X} \overset{d}{=} \underline{X}' \Rightarrow A = A', B = B', C = C'.$$

*or equivalently for $\mathfrak{C}(M, Q)$ and $\mathfrak{C}'(M', Q')$,*

$$\underline{X} \overset{d}{=} \underline{X}' \Rightarrow M = M', Q = Q',$$

Correspondingly, we say that SCMs $\mathfrak{C}(A, B, C)$ and $\mathfrak{C}'(A', B', C')$ are distinct if $(A, B, C) \neq (A', B', C')$ or equivalently, $(M, Q) \neq (M', Q')$.

**Definition 5 (Dependent random vector)** *A random vector $\underline{Y}$ is* independent *if its entries are mutually independent; otherwise, $\underline{Y}$ is* dependent.

Refer to Definition 13 and Remarks 5 and 6 in the Appendix for the definition of the characteristic function (CF) and its fundamental properties.

**Definition 6 (Decomposable characteristic function)** *(Lukacs, 1972) A CF is decomposable if it can be written as the product of two CF's such that neither take a degenerate distribution. A degenerate distribution is a distribution of a constant number with probability one.*

## 3 LINEAR SCM WITH GAUSSIAN CONFOUNDERS

The SCM defined in (1) with Gaussian confounders is not finitely identifiable, because the noise variables are also Gaussian. Specifically, each noise variable $Z_i$ (for $i = 1, \ldots, n$) can be expressed as the sum of another noise component $Z_i'$ and a confounder $H_i'$, where $H_i'$ affects only the observed variable $X_i$. This decomposition satisfies $E[Z_i^2] = E[Z_i'^2] + E[H_i'^2]$. Since $Z_i'$ and $H_i'$ can have different variances, there are infinitely many SCMs corresponding to different choices of these variances. To avoid addressing these trivial cases and provide a meaningful distinction between noise and confounding variables, we make the following assumption.

**Assumption 1** *Each confounder causes at least two observed variables.*

The assumption ensures that every column of matrix $B$ has at least two nonzero entries. The following theorem shows that even under this assumption, the SCM is not finitely identifiable.

**Theorem 1** *Under Assumption 1, SCM $\mathfrak{C}$ defined in* (1) *with Gaussian confounders is not finitely identifiable if and only if the observed vector is dependent. Moreover, if the observed vector is independent, then SCM $\mathfrak{C}$ is uniquely identifiable.*

**Proposition 1** *SCM $\mathfrak{C}$ defined in* (1) *with Gaussian confounders and a dependent observed vector can be modeled as another SCM without confounders that has the same PDF for the observed variables and vice versa.*

Proposition 1 holds even if SCM $\mathfrak{C}$ satisfies Assumption 1.

As discussed earlier, Assumption 1 is not restrictive for the sufficiency part of Theorem 1 as each noise can be decomposed into another noise and a confounder component. This result holds even if the number of confounders are restricted to be less than the number of observed variables, which is often assumed in practice.

**Proposition 2** *SCM $\mathfrak{C}$ defined in* (1) *with Gaussian confounders and a dependent observed vector is not finitely identifiable. The result remains valid if the number of confounders is known to be less than the number of observed variables.*

## 4 LINEAR SCM WITH NON-GAUSSIAN CONFOUNDERS

We present the sufficient conditions for finite identifiability of the SCMs when the confounders satisfy the following assumption.

**Assumption 2** *Confounders $H_i, i = 1, \ldots, k$, are non-constant, non-Gaussian, and cannot be decomposed into finite random variables where any is Gaussian. In other words, for any $Y$ and $\tilde{Y}$ for which $\phi_{H_i}(t) = \phi_Y(t)\phi_{\tilde{Y}}(t)$, neither $Y$ nor $\tilde{Y}$ is Gaussian.*

The enumeration of the confounders can be done arbitrarily, as for any enumeration, the columns of matrix $B$ (resp. $M$) can be reordered to maintain the consistency of the product $B\underline{H}$ (resp. $M\underline{H}$), which encapsulates the confounders' impact on the observed variables. Therefore, to avoid complications with confounder enumeration, we assume the following without loss of generality.

**Assumption 3** *The columns of $M$ are sorted in the ascending lexicographical order.*

**Assumption 4** *The columns of $M$ in* (2) *are linearly independent.*

Intuitively, this assumption means that the confounders' "impacts" on the observed variables are independent, and hence, distinguishable. The following is the main result of this section.

**Theorem 2** *Under Assumptions 2 to 4, an $n$-dimensional linear SCM $\mathfrak{C}(M, Q)$ defined by* (2) *with $k$ confounders, $1 \leq k \leq n$,*

    *1. is finitely identifiable, and the size of the equivalence class is at most $n!$,*

2. *for every SCM $\mathfrak{C}'(M', Q')$ in this class, it holds that $M' = M$ and $Q' = P^\top Q_p P$ for some permutation matrix $P$ where $Q_p$ is the lower triangular matrix obtained from the Cholesky decomposition of $PQQ^\top P^\top$, and*

3. *the CF of the confounder vector is the same in all SCMs in the equivalence class and is given by*

$$\phi_{\underline{H}}(\underline{t}) = \frac{\phi_{\underline{\tilde{X}}}\big((\tilde{M}^{-1})^\top \underline{t}\big)}{\phi_{\underline{N}}\big(\tilde{Q}^\top (\tilde{M}^{-1})^\top \underline{t}\big)} \tag{3}$$

*where $\phi_{\underline{N}}(\underline{t})$ denotes the CF of a standard $k$-dimensional Gaussian random variable, and $\underline{\tilde{X}}$ is a vector of informative observed variables, whose corresponding rows in $M$ and $Q$ form the matrices $\tilde{M}$ and $\tilde{Q}$, respectively.*

As described in Theorem 2, each matrix $P$ corresponds to an SCM $\mathfrak{C}'$ that is equivalent to the SCM $\mathfrak{C}$. In what follows, we examine how matrix $P$ affects the causal structure of SCM $\mathfrak{C}'$. To this aim, consider the following definitions. Due to the uniqueness of the Cholesky decomposition, each permutation matrix $P$ in Theorem 2 corresponds to exactly one SCM $\mathfrak{C}'$. This uniqueness guarantees the existence of a function that maps $P$ to $\mathfrak{C}'$.

**Definition 7** *Consider an SCM $\mathfrak{C}(M, Q)$ with $k$ confounders, where $1 \leq k \leq n$. Let Assumptions 2 to 4 hold. Define the function $\mathcal{F}_{\mathfrak{C}}$ that maps a permutation matrix $P$ to its corresponding SCM $\mathfrak{C}'$ in the finite equivalence class of $\mathfrak{C}$, as described in Theorem 2–Part 2, i.e., $\mathcal{F}_{\mathfrak{C}}(P) = \mathfrak{C}'$, where $\mathfrak{C}'(M, P^\top Q_p P)$ with $Q_p$ being the lower triangular matrix obtained from the Cholesky decomposition of $PQQ^\top P^\top$.*

**Definition 8** *The "order" of entries $X_1, \ldots, X_n$ in a vector $\underline{X} = [X_1, \ldots, X_n]^\top$ is $X_1 \succ \ldots \succ X_n$. The "permutation order" $\mathcal{O}_P(\underline{X})$ for an $n \times n$ permutation matrix $P$ represents the order of variables $\underline{X}$ in $P\underline{X}$.*

The following theorem presents the conditions under which the permutation matrix $P$ leads to a new SCM $\mathfrak{C}'$ and subsequently shows how $P$ affects the causal order of $\mathfrak{C}'$.

**Theorem 3** *Consider SCM $\mathfrak{C}(M, Q)$ defined by (2) with $k$ confounders, $1 \leq k \leq n$, and let Assumptions 2 to 4 hold. Let $P$ be a permutation matrix with permutation order $\mathcal{O}_P(\underline{X})$ and $\mathcal{F}_{\mathfrak{C}}(P) = \mathfrak{C}'$. Then,*

1. *$\mathfrak{C}$ and $\mathfrak{C}'$ are distinct SCMs if and only if $\mathcal{O}_P(\underline{X}) \notin \mathfrak{D}_{\mathfrak{C}}$; and*

2. *the permutation order $\mathcal{O}_P(\underline{X})$ is an element of the set $\mathfrak{D}_{\mathfrak{C}'}$.*

For the proof of part (1), we need the following lemma.

**Lemma 1** *Consider SCM $\mathfrak{C}(M, Q)$ defined by (2) with $k$ confounders, $1 \leq k \leq n$, and let Assumptions 2 to 4 hold. Let $P$ be a permutation matrix with permutation order $\mathcal{O}_P(\underline{X})$ and $\mathcal{F}_{\mathfrak{C}}(P) = \mathfrak{C}'$. Then, $\mathfrak{C}(M, Q)$ and $\mathfrak{C}'(M, Q')$ are distinct SCMs (i.e., $Q \neq Q'$) if and only if $PQP^\top$ is a lower triangular matrix.*

**Corollary 1** *SCM $\mathfrak{C}(M, Q)$ defined by (2) satisfying Assumptions 2 to 4 and with $k$ confounders, $1 \leq k \leq n$, is uniquely identifiable if the causality order is specified.*

**Definition 9 (Partial causal (structure) order)** *A partial causal order for SCM $\mathfrak{C}$ is an order $\mathcal{X}_1 \succ \ldots \mathcal{X}_m$ where $\mathcal{X}_1, \ldots, \mathcal{X}_m$ partition the observed variables $\{X_1, \ldots, X_n\}$, and there is no path from any node in $\mathcal{X}_i$ to a node in $\mathcal{X}_j$ in the induced DAG for every $i, j \in \{1, 2, \ldots, m\}, i > j$.*

**Corollary 2** *Consider SCM $\mathfrak{C}(M, Q)$ defined by (2) satisfying Assumptions 2 to 4 with $k$ confounders, $1 \leq k \leq n$. If a partial causal order $\mathcal{X}_1 \succ \ldots \succ \mathcal{X}_m$ is specified, then the size of the equivalence class of $\mathfrak{C}$ is at most $\prod_{i=1}^{m} |\mathcal{X}_i|!$.*

Define an isolated observed variable as an observed variable that is not caused by any other variable.

**Definition 10 (Isolated observed variable)** *Given SCM* (1)*, an observed variable $X_i$ is said to be isolated if the $i^{th}$ row and $i^{th}$ column of $\boldsymbol{A}$ are zero.*

The next corollary implies that an SCM is uniquely identifiable if all observed variables are influenced only by unobserved variables, i.e., confounders and noise.

**Corollary 3** *SCM $\mathfrak{C}(\boldsymbol{M}, \boldsymbol{Q})$ defined by (2) satisfying Assumptions 2 to 4 and with $k$ confounders, $1 \leq k \leq n$, is uniquely identifiable if and only if all observed variables are isolated.*

**Definition 11 (Isolated variable set)** *Given SCM* (1)*, an observed variable set $\{X_{i_1}, \ldots, X_{i_m}\}$ is said to be isolated if $A_{ab} = A_{ba} = 0$ for all $a \in \{i_1, \ldots, i_m\}$ and $b \notin \{i_1, \ldots, i_m\}$.*

**Corollary 4** *Consider SCM $\mathfrak{C}(\boldsymbol{M}, \boldsymbol{Q})$ defined by (2) satisfying Assumptions 2 to 4 and with $k$ confounders, $1 \leq k \leq n$. If $\mathfrak{C}$ consists of isolated variable sets $\mathcal{X}_1, \ldots, \mathcal{X}_m$, then so is every other SCM in the equivalence class of $\mathfrak{C}$. Moreover, the size of the equivalence class is at most $\prod_{i=1}^{m} |\mathcal{X}_i|!$.*

# 5 COMPARISON OF IDENTIFIABILITY RESULTS FOR DIFFERENT NOISE AND CONFOUNDER DISTRIBUTIONS

Building on the findings of this article and (Pavan & Miranda, 2018), in the following remarks, we compare the identifiability of SCM $\mathfrak{C}(\boldsymbol{M}, \boldsymbol{Q})$ in different scenarios, depending on whether noise and confounders follow a Gaussian distribution.

**Remark 2 (Unique identifiability)** *Consider SCM $\mathfrak{C}(\boldsymbol{M}, \boldsymbol{Q})$ with equations defined in 2 and with $k$ confounders. Then*

- ***Non-Gaussian noise and confounders:*** *SCM $\mathfrak{C}$ is identifiable up to the permutations of the matrix $[\boldsymbol{M}, \boldsymbol{Q}]$ (Pavan & Miranda, 2018), i.e., $\mathfrak{C}$ is finitely identifiable. Unlike in (Pavan & Miranda, 2018), the scaling of $[\boldsymbol{M}, \boldsymbol{Q}]$ changes the observed PDF, because the covariance matrices are assumed to equal the identity matrix. Also, permutations are acceptable only if they preserve the structure of the $\boldsymbol{Q}$ matrix, specifically, $Q_{ij} \cdot Q_{ji} = 0$ and $Q_{ii} \neq 0$ for $i, j \in 1, 2, \ldots, n, i \neq j$. Now if all "accepteble" permutation matrices result in the same $[\boldsymbol{M}, \boldsymbol{Q}]$, SCM $\mathfrak{C}$ is uniquely identifiable.*
- ***Gaussian noise and non-Gaussian confounders:*** *SCM $\mathfrak{C}$ is finitely identifiable under Assumptions 2 to 4 and $1 \leq k \leq n$. All observed variables being isolated is the necessary and sufficient condition for unique identifiability.*
- ***Gaussian noise and confounders:*** *SCM $\mathfrak{C}$ is not finitely identifiable. Under Assumption 1, observed variables are mutually independent is the necessary and sufficient condition for unique identifiability.*

Suppose the causal order is known, for example, through expert knowledge. Would the SCM then be identifiable? The following remark answers this question.

**Remark 3 (Unique identifiability given the causal order)** *Consider SCM $\mathfrak{C}(\boldsymbol{M}, \boldsymbol{Q})$ with equations defined in 2 and with $k$ confounders. Assume that the causal order of SCM $\mathfrak{C}$ is known. Then*

- ***Non-Gaussian noise and confounders:*** *SCM $\mathfrak{C}$ may or may not be uniquely identifiable;*
- ***Gaussian noise and non-Gaussian confounders:*** *SCM $\mathfrak{C}$ is uniquely identifiable;*
- ***Gaussian noise and confounders:*** *SCM $\mathfrak{C}$ is not finitely identifiable for dependent observed variables.*

In certain cases, even if the SCM itself is not identifiable, determining the causal order can still be of value. The following remark outlines the conditions under which this may be possible.

**Remark 4 (Identifiability of the causal order)** *Consider SCM $\mathfrak{C}(\boldsymbol{M}, \boldsymbol{Q})$ with equations defined in 2 and with $k$ confounders. Consider the case where SCM $\mathfrak{C}$ is not uniquely identifiable. Then for*

- ***Non-Gaussian noise and confounders:*** *The causal order of SCM $\mathfrak{C}$ may or may not be unique in its equivalence class;*

- **Gaussian noise and non-Gaussian confounders:** *The causal order of SCM $\mathfrak{C}$ is not unique in its equivalence class;*
- **Gaussian noise and confounders:** *The causal order of SCM $\mathfrak{C}$ is not unique in its equivalence class.*

The third parts of Remarks 3 and 4 are proven in the appendix. The first parts of the remarks, i.e., non-Gaussian noise and confounders, are illustrated by the following SCMs:

$$[\boldsymbol{M_1}, \boldsymbol{Q_1}] = \begin{bmatrix} 1 & 1 & 0 \\ 1 & 1 & 1 \end{bmatrix}, \, [\boldsymbol{M_2}, \boldsymbol{Q_2}] = \begin{bmatrix} 1 & 1 & 0 & 0 \\ 0 & 1 & 1 & 0 \\ 1 & 1 & 1 & 1 \end{bmatrix}, \, [\boldsymbol{M_3}, \boldsymbol{Q_3}] = \begin{bmatrix} 1 & 1 & 0 & 0 \\ 0 & 1 & 1 & 0 \\ 1 & 1 & 1 & 1 \end{bmatrix}, \, [\boldsymbol{M_4}, \boldsymbol{Q_4}] = \begin{bmatrix} 0 & 1 & 1 & 0 \\ 1 & 0 & 1 & 0 \\ 1 & 1 & 1 & 1 \end{bmatrix}.$$

There is no acceptable permutation for $[\boldsymbol{M_1}, \boldsymbol{Q_1}]$ to result in a new matrix $[\boldsymbol{M}, \boldsymbol{Q}]$, implying that the corresponding SCM is uniquely identifiable. For $[\boldsymbol{M_2}, \boldsymbol{Q_2}]$, the only acceptable permutation is to swap the first and second columns, which leads to a new SCM but with the same causal order. This demonstrates two different SCMs with the same PDF and causal order. In the case of $[\boldsymbol{M_3}, \boldsymbol{Q_3}]$, the causal order is $X_1 \succ X_2 \succ X_3$. Permuting it to $[\boldsymbol{M_4}, \boldsymbol{Q_4}]$ is acceptable and results in an SCM with a new causal order $X_2 \succ X_1 \succ X_3$.

## 6 FINDING EQUIVALENCE CLASS AND NUMERICAL EXAMPLES

The following algorithm illustrates how an equivalence class is obtained for a specified SCM $\mathfrak{C}(\boldsymbol{M}, \boldsymbol{Q})$ that satisfies Assumptions 2–4. The computational complexity is dominated by operations on $n \times n$ matrices. The most intensive steps—matrix multiplication ($\boldsymbol{\Sigma} = \boldsymbol{Q}\boldsymbol{Q}^\top$), Cholesky decomposition of $\boldsymbol{\Sigma}_p$, and inversion of $\boldsymbol{Q}'$—each require $O(n^3)$ time. Thus, the overall complexity is $O(n^3)$.

---

**Algorithm 1:** Finding the SCM equivalent to SCM $\mathfrak{C}$ and corresponding to permutation matrix $\boldsymbol{P}$

---

**Require:** matrices $\boldsymbol{Q}$, $\boldsymbol{M}$, and $\boldsymbol{P}$

   $\boldsymbol{M}' \leftarrow \boldsymbol{M}$
   $\boldsymbol{\Sigma} \leftarrow \boldsymbol{Q}\boldsymbol{Q}^\top$
   $\boldsymbol{\Sigma}_p \leftarrow \boldsymbol{P}\boldsymbol{\Sigma}\boldsymbol{P}^\top$
   $\boldsymbol{\Sigma}_p^L \leftarrow$ Cholesky decomposition$(\boldsymbol{\Sigma}_P)$
   $\boldsymbol{Q}' \leftarrow \boldsymbol{P}^\top \boldsymbol{\Sigma}_p^L \boldsymbol{P}$
   **if** $\boldsymbol{Q}' = \boldsymbol{Q}$ **then**
      **print** This $\boldsymbol{P}$ does not result in a distinct SCM.
   **else**
      $\boldsymbol{C}' \leftarrow \boldsymbol{0}_{n \times n}$
      **for** $i = 1 : n$ **do**
         $C'_{ii} \leftarrow Q'_{ii}$
      **end for**
      $\boldsymbol{A}' \leftarrow \boldsymbol{I_n} - \boldsymbol{C}'\boldsymbol{Q}'^{-1}$
      $\boldsymbol{B}' \leftarrow (\boldsymbol{I_n} - \boldsymbol{A}')\boldsymbol{M}'$
      **return** $\boldsymbol{A}', \boldsymbol{B}', \boldsymbol{C}'$
   **end if**

---

In the following, we present two examples. First, we discuss a 2-dimensional SCM, also presented in (Hoyer et al., 2008; Salehkaleybar et al., 2020), and illustrate the identifiability of the SCM under different distributions for the noise and confounders.

**Example 1** *Consider the SCM defined by*

$$\mathfrak{C} : \begin{cases} X_1 = H + \sqrt{2}Z_1, \\ X_2 = H + X_1 + Z_2, \end{cases}$$

*where $E[X_1^2] = 3$, $E[X_1 X_2] = 4$, and $E[X_2]^2 = 7$, We examine this SCM in three different scenarios to determine whether there are other SCMs with the same PDF.*

**(1) All variables are non-Gaussian**: *Then under the assumption that $H' = Z_1$, $Z_1' = H$, and $Z_2 = Z_2'$, the following SCM has the same joint PDF over the observed variables:*

$$\mathfrak{C}' : \begin{cases} X_1' = \sqrt{2}H' + Z_1', \\ X_2' = -\sqrt{2}H' + 2X_1' + Z_2'. \end{cases}$$

**(2) All variables are Gaussian**: *There are infinite SCMs $\mathfrak{C}'$ with the same joint PDF, defined as*

$$\mathfrak{C}' : \begin{cases} X_1' = b_1 H' + a_{12} X_2' + c_1 Z_1', \\ X_2' = b_2 H' + a_{21} X_1' + c_2 Z_2', \end{cases}$$

*where $a_{12}.a_{21} = 0$. For $\mathbb{P}(\underline{X}') = \mathbb{P}(\underline{X})$ to hold, it suffices that either $a_{12} = 0$, $|b_1| < \sqrt{3}$, $3a_{21} + b_2 b_1 = 4$, and $b_2^2 + 3a_{21}^2 + 2a_{21} b_1 b_2 < 7$ or $a_{21} = 0$, $|b_2| < \sqrt{7}$, $7a_{12} + b_2 b_1 = 4$, and $b_1^2 + 7a_{12}^2 + 2a_{12} a_1 b_2 < 3$. It can be verified that there exist infinitely many parameter values that satisfy these conditions, implying that there are infinitely many SCMs, confirming Theorem 1. Moreover, setting $a_{12} = b_1 = b_2 = 0$, $a_{21} = {}^4\!/\!3$, results in $c_1^2 = 3$ and $c_2^2 = {}^5\!/\!3$. That is, an SCM without confounders can also be obtained, which is consistent with Proposition 1.*

**(3) Gaussian noise and non-Gaussian confounders:** *Let Assumption 2 be in force. Then in view of Theorem 2, there is at most one other SCM with the same joint PDF, which has a different causal order $X_2 \succ X_1$, and equals the following (which can be obtained from Algorithm 1):*

$$\mathfrak{C}' = \begin{cases} X_1' = -{}^1\!/\!3 H' + {}^2\!/\!3 X_2' + \sqrt{2/3} Z_1', \\ X_2' = 2H' + \sqrt{3} Z_2'. \end{cases}$$

The next example is a 3-dimensional SCM satisfying Assumptions 2–4. The example illustrates given an SCM $\mathfrak{C}$, when a permutation matrix $\boldsymbol{P}$ results in a distinct SCM $\mathfrak{C}'$ in the equivalence class of $\mathfrak{C}$.

**Example 2** *Consider the following SCM:*

$$\mathfrak{C} : \underline{X} = \begin{bmatrix} 0 & 0 & 0 \\ 1 & 0 & 0 \\ 1 & 0 & 0 \end{bmatrix} \underline{X} + \begin{bmatrix} 1 & 1 & 1 \\ 1 & 1 & 2 \\ -1 & 2 & 1 \end{bmatrix} \underline{H} + \begin{bmatrix} 2 & 0 & 0 \\ 0 & 1 & 0 \\ 0 & 0 & \sqrt{3} \end{bmatrix} \underline{Z}, \tag{4}$$

$$\boldsymbol{M} = (\boldsymbol{I}_3 - \boldsymbol{A})^{-1}\boldsymbol{B} = \begin{bmatrix} 1 & 1 & 1 \\ 2 & 2 & 3 \\ 0 & 3 & 2 \end{bmatrix}, \ \boldsymbol{Q} = (\boldsymbol{I}_3 - \boldsymbol{A})^{-1}\boldsymbol{C} = \begin{bmatrix} 2 & 0 & 0 \\ 2 & 1 & 0 \\ 2 & 0 & \sqrt{3} \end{bmatrix}, \ \boldsymbol{\Sigma} = \boldsymbol{Q}\boldsymbol{Q}^\top = \begin{bmatrix} 4 & 4 & 4 \\ 4 & 5 & 4 \\ 4 & 4 & 7 \end{bmatrix}.$$

*It follows that $\mathfrak{D}_c = \{X_1 \succ X_2 \succ X_3, X_1 \succ X_3 \succ X_2\}$. Clearly, Assumptions 3 and 4 hold. Let Assumption 2 to also be in force. Since $n = 3$, there are six permutation matrices corresponding to different causal orders:*

$$\boldsymbol{P}_1 = \begin{bmatrix} 1 & 0 & 0 \\ 0 & 1 & 0 \\ 0 & 0 & 1 \end{bmatrix}, \boldsymbol{P}_2 = \begin{bmatrix} 0 & 1 & 0 \\ 1 & 0 & 0 \\ 0 & 0 & 1 \end{bmatrix}, \boldsymbol{P}_3 = \begin{bmatrix} 0 & 0 & 1 \\ 0 & 1 & 0 \\ 1 & 0 & 0 \end{bmatrix}, \boldsymbol{P}_4 = \begin{bmatrix} 1 & 0 & 0 \\ 0 & 0 & 1 \\ 0 & 1 & 0 \end{bmatrix}, \boldsymbol{P}_5 = \begin{bmatrix} 0 & 0 & 1 \\ 1 & 0 & 0 \\ 0 & 1 & 0 \end{bmatrix}, \boldsymbol{P}_6 = \begin{bmatrix} 0 & 1 & 0 \\ 0 & 0 & 1 \\ 1 & 0 & 0 \end{bmatrix}$$

*By applying Algorithm 1 to each permutation matrix, the following results are obtained. The numbers are rounded to two decimal places:*

*(1) $\boldsymbol{P}_1$: According to Proposition 3, this permutation matrix implies the permutation order $\mathcal{O}_{\boldsymbol{P}_1}$ : $X_1 \succ X_2 \succ X_3$. The algorithm outputs the same SCM as in (4). This consists with Theorem 3 as $\mathcal{O}_{\boldsymbol{P}_1} \in \mathfrak{D}_{\mathfrak{C}}$.*

*(2) $\boldsymbol{P}_2$ : The corresponding permutation order is $\mathcal{O}_{\boldsymbol{P}_2}$ : $X_2 \succ X_1 \succ X_3$. Thus, $\mathcal{O}_{\boldsymbol{P}_2} \notin \mathfrak{D}_{\mathfrak{C}}$. The algorithm outputs the following distinct SCM:*

$$\mathfrak{C}_1' : \underline{X}' = \begin{bmatrix} 0 & 0.8 & 0 \\ 0 & 0 & 0 \\ 1 & 0 & 0 \end{bmatrix} \underline{X}' + \begin{bmatrix} -0.6 & -0.6 & 1.4 \\ 2 & 2 & 3 \\ -1 & 2 & 1 \end{bmatrix} \underline{H}' + \begin{bmatrix} 0.89 & 0 & 0 \\ 0 & 2.24 & 0 \\ 0 & 0 & 1.73 \end{bmatrix} \underline{Z}'.$$

*(3) $\boldsymbol{P}_3$ : The corresponding permutation order is $\mathcal{O}_{\boldsymbol{P}_3}$ : $X_3 \succ X_2 \succ X_1$. Thus, $\mathcal{O}_{\boldsymbol{P}_3} \notin \mathfrak{D}_{\mathfrak{C}}$. The algorithm outputs the following distinct SCM:*

$$\mathfrak{C}_2' : \underline{X}' = \begin{bmatrix} 0 & 0.63 & 0.21 \\ 0 & 0 & 0.57 \\ 0 & 0 & 0 \end{bmatrix} \underline{X}' + \begin{bmatrix} -0.26 & -0.89 & 1.32 \\ 2 & 0.29 & 1.85 \\ 0 & 3 & 2 \end{bmatrix} \underline{H}' + \begin{bmatrix} 0.79 & 0 & 0 \\ 0 & 1.65 & 0 \\ 0 & 0 & 2.65 \end{bmatrix} \underline{Z}'.$$

*(4) $\boldsymbol{P}_4$ : The corresponding permutation order is $\mathcal{O}_{\boldsymbol{P}_4}$ : $X_1 \succ X_3 \succ X_2$. This, $\mathcal{O}_{\boldsymbol{P}_4} \in \mathfrak{D}_{\mathfrak{C}}$, which does not result in a distinct SCM.*

*(5)* $\boldsymbol{P_5}$ *: The corresponding permutation order is* $X_3 \succ X_1 \succ X_2$. *Thus,* $\mathcal{O}_{\boldsymbol{P_5}} \notin \mathfrak{O}_{\mathfrak{C}}$. *The algorithm outputs the following distinct SCM:*

$$\mathfrak{C}'_3 : \underline{X}' = \begin{bmatrix} 0 & 0 & 0.57 \\ 1 & 0 & 0 \\ 0 & 0 & 0 \end{bmatrix} \underline{X}' + \begin{bmatrix} 1 & -0.71 & -0.14 \\ 1 & 1 & 2 \\ 0 & 3 & 2 \end{bmatrix} \underline{H}' + \begin{bmatrix} 1.31 & 0 & 0 \\ 0 & 1 & 0 \\ 0 & 0 & 2.65 \end{bmatrix} \underline{Z}'.$$

*(6)* $\boldsymbol{P_6}$ *: The corresponding permutation order is* $X_2 \succ X_3 \succ X_1$. *Thus,* $\mathcal{O}_{\boldsymbol{P_6}} \notin \mathfrak{O}_{\mathfrak{C}}$. *The algorithm outputs the following distinct SCM:*

$$\mathfrak{C}'_4 : \underline{X}' = \begin{bmatrix} 0 & 0.63 & 0.21 \\ 0 & 0 & 0 \\ 0 & 0.8 & 0 \end{bmatrix} \underline{X}' + \begin{bmatrix} -0.26 & -0.89 & -1.32 \\ 2 & 2 & 3 \\ -1.6 & 1.4 & -0.4 \end{bmatrix} \underline{H}' + \begin{bmatrix} 0.79 & 0 & 0 \\ 0 & 2.24 & 0 \\ 0 & 0 & 1.95 \end{bmatrix} \underline{Z}'.$$

*Therefore, the 3-dimensional SCM in this example is identifiable with an equivalence class size of five.*

## 7 CONCLUSION

We studied the identifiability of SCMs from observational data, i.e., sampled from $\mathbb{P}(\underline{X})$, focusing on scenarios with Gaussian and non-Gaussian noise and confounders. When both noise and confounders are Gaussian, it is impossible to determine the causal order solely from observational data. This scenario often results in the existence of infinitely many SCMs with the same causal order and same PDF of observed variables $\mathbb{P}(\underline{X})$, confirming that Gaussian SCMs are not finitely identifiable.

In contrast, in scenarios where all variables are non-Gaussian, it was already known from the literature that the SCM is identifiable up to permutations of the structural matrices $[\boldsymbol{M}, \boldsymbol{Q}]$ (Pavan & Miranda, 2018; Eriksson & Koivunen, 2004; Tharwat, 2021), resulting in finite identifiability. Depending on the structure of $[\boldsymbol{M}, \boldsymbol{Q}]$, the equivalence class may either include a unique SCM, multiple SCMs with different causal orders, or multiple SCMs sharing the same causal order.

For Gaussian noise and non-Gaussian confounders, we showed that the SCM is finitely identifiable. More importantly, the SCM is uniquely identifiable once the causal order is known; however, determining the causal order is not possible unless the observed variables are isolated.

Looking ahead, we acknowledge several assumptions that restrict the scope of this study, such as the SCM being linear and the confounders lack Gaussian components. Additionally, while we discussed deriving other SCMs from a known SCM, we did not show how to estimate that "first" SCM from observational data. Future research should aim to relax these constraints to investigate more realistic scenarios.

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

## A APPENDIX

**Definition 12 (Equal in distribution)** *Two real random variables $\underline{X}$ and $\underline{Y}$ with the same state space are equal in the distribution sense, denoted by $\underline{X} \stackrel{d}{=} \underline{Y}$, if for any measurable set $\mathcal{A}$ in the state space, $P(\underline{X} \in \mathcal{A}) = P(\underline{Y} \in \mathcal{A})$.*

**Definition 13 (Characteristic function)** *Define $\phi_{\underline{X}}(t)$ the* Characteristic Function (CF) *of $(X_1, X_2, \ldots, X_n)$ as*

$$\phi_{(X_1, X_2, \ldots, X_n)}(t_1, t_2, \ldots, t_n) = E_{(X_1, X_2, \ldots, X_n)} \left[ \exp \left( j(X_1 t_1 + X_2 t_2 + \ldots + X_n t_n) \right) \right] \quad (5)$$

*where $j^2 = -1$.*

**Remark 5** *The CF of $m$ independent random variables $Y_1, \ldots, Y_m$ can be obtained as the product of each random variable CF. $\phi_{\underline{Y}}(\underline{t}) = \prod_{i=1}^{m} \phi_{Y_i}(t_i)$.*

**Remark 6** *The CF of the sum of independent random variables $Y_1, \ldots, Y_m$ can be obtained from the multiplication of their CF's. Therefore, the CF of $S = \sum_{i=1}^{m} Y_i$ is obtained as $\phi_S(t) = \prod_{i=1}^{m} \phi_{Y_i}(t)$.*

**Lemma 2** *For random vectors $\underline{X}$ and $\underline{Y} = \boldsymbol{A}\underline{X} + \underline{b}$ (where $\boldsymbol{A}$ is a constant matrix and $\underline{b}$ a constant vector), $\Phi_{\underline{Y}}(\underline{t}) = \exp(j\underline{b}^\top\underline{t})\Phi_{\underline{X}}(\boldsymbol{A}^\top\underline{t})$ where $j^2 = -1$ (Andersen & Sorensen, 1995).*

**Lemma 3** *Every positive definite matrix can be written as the sum of another positive matrix and a diagonal positive definite matrix.*

**Proof.** Consider a positive definite matrix $\boldsymbol{\Sigma}$, and denote its eigenvalues be denoted by $\lambda_i(\boldsymbol{\Sigma})$ which are all positive. Choose a positive number $d$ such that

$$d < \min_{i=1,2,\ldots,n} \lambda_i(\boldsymbol{\Sigma}).$$

Form the diagonal matrix $\boldsymbol{D}$ with entries $d$ on the diagonal. Define $\boldsymbol{S} = \boldsymbol{\Sigma} - \boldsymbol{D}$. We show that $\boldsymbol{S}$ is positive definite. First, since $\boldsymbol{\Sigma}$ is symmetric and $\boldsymbol{D}$ is diagonal, $\boldsymbol{S}$ is also symmetric. To show that $\boldsymbol{S}$ is positive definite, consider its eigenvalues. Second, the eigenvalues of $\boldsymbol{S}$ are $\lambda_i(\boldsymbol{\Sigma}) - d$ (Horn & Johnson, 2012). Since $d < \min_{i=1,2,\ldots,n} \lambda_i(\boldsymbol{\Sigma})$, it follows that $\lambda_i(\boldsymbol{S}) > 0$ for all $i \in \{1, 2, \ldots, n\}$. Thus, $\boldsymbol{S}$ is a positive definite matrix. $\square$

**Lemma 4** *Let $\boldsymbol{\mathcal{L}}$ be the Cholesky factor of a positive definite matrix $\boldsymbol{S}$. Define matrix $\boldsymbol{B}$ as the collection of those columns of $\boldsymbol{\mathcal{L}}$ that have more than one non-zero element, in an arbitrary order. Then, $\tilde{\boldsymbol{S}} = \boldsymbol{S} - \boldsymbol{B}\boldsymbol{B}^\top$ is a non-negative diagonal matrix.*

**Proof.** Let $\boldsymbol{\mathcal{L}} = [\underline{L}_1, \underline{L}_2, \ldots, \underline{L}_n]$, where $\underline{L}_i$ denotes the $i^{\text{th}}$ column. Then $\boldsymbol{\mathcal{L}}\boldsymbol{\mathcal{L}}^\top = \sum_{i=1}^n \underline{L}_i\underline{L}_i^\top$. Since $\boldsymbol{S}$ is positive definite, all diagonal entries of $\boldsymbol{\mathcal{L}}$ are non-zero. Now, let $\mathcal{T}$ denote the indices of the columns with more than one non-zero elements. Then $\boldsymbol{B}\boldsymbol{B}^\top = \sum_{i\in\mathcal{T}} \underline{L}_i\underline{L}_i^\top$. Consequently, $\tilde{\boldsymbol{S}} = \boldsymbol{\mathcal{L}}\boldsymbol{\mathcal{L}}^\top - \boldsymbol{B}\boldsymbol{B}^\top = \sum_{j\notin\mathcal{T}} \underline{L}_j\underline{L}_j^\top$. Since each column $\underline{L}_j$, $j \notin \mathcal{T}$, has exactly one non-zero entry, located at the $j^{\text{th}}$ row, $\underline{L}_j\underline{L}_j^\top$ is a matrix with all zero entries except for the $j^{\text{th}}$ column and row, which is a positive entry. Thus, the summation of $\underline{L}_j\underline{L}_j^\top$, $j \notin \mathcal{T}$, is a non-negative diagonal matrix. $\square$

**Proof of Theorem 1:** Consider the covariance matrix of the observed vector, denoted by $\boldsymbol{\Sigma} = E[\underline{X}\,\underline{X}^\top]$. Being positive definite, $\boldsymbol{\Sigma}$ can be decomposed as $\boldsymbol{\Sigma} = \boldsymbol{D} + \boldsymbol{S}$, where $\boldsymbol{D}$ is a positive definite diagonal matrix and $\boldsymbol{S}$ is a positive definite matrix, according to Lemma 3.

(sufficiency) The dependent observed vector implies that $\boldsymbol{\Sigma}$ is not diagonal. Thus, $\boldsymbol{S}$ cannot be diagonal. Consider an arbitrary $\epsilon \in \mathbb{R}$ that meets two conditions: *(i)* $0 < \epsilon < \min_{i=1,\ldots,n}(\boldsymbol{D}_{ii})$ and *(ii)* $\boldsymbol{S} + \epsilon\boldsymbol{I}_n$ is not a diagonal matrix. Define SCM $\mathfrak{C}^\epsilon$ as

$$\underline{X}^\epsilon = \boldsymbol{B}^\epsilon\underline{H}^\epsilon + \boldsymbol{C}^\epsilon\underline{Z}^\epsilon.$$

where $\boldsymbol{C}^\epsilon$ and $\boldsymbol{B}^\epsilon$ are defined as follows. Let $\boldsymbol{\mathcal{L}}^\epsilon$ denote the Cholesky factor of $\boldsymbol{S} + \epsilon\boldsymbol{I}_n$. The matrix $\boldsymbol{B}^\epsilon$ is formed by selecting all columns of $\boldsymbol{\mathcal{L}}^\epsilon$ that have at least two non-zero elements. Let $m$ be the number of such columns in $\boldsymbol{\mathcal{L}}^\epsilon$, implying that each confounder causes at least two observed variables in $\mathfrak{C}^\epsilon$. Then the dimension of $\boldsymbol{B}^\epsilon$ is $n \times m$ where $m$ is not necessarily the same as $k$, the number of confounders in $\mathfrak{C}$. Due to non-diagonality of $\boldsymbol{S} + \epsilon\boldsymbol{I}_n$, it follows that $1 \leq m$, which guarantees the existence of $\boldsymbol{B}^\epsilon$. Define $\tilde{\boldsymbol{S}}^\epsilon = \boldsymbol{S} + \epsilon\boldsymbol{I}_n - \boldsymbol{B}^\epsilon(\boldsymbol{B}^\epsilon)^\top$. It is easy to verify that $\tilde{\boldsymbol{S}}^\epsilon$ is a diagonal matrix with non-negative entries (Lemma 4). Therefore, the matrix $\boldsymbol{D} + \tilde{\boldsymbol{S}}^\epsilon - \epsilon\boldsymbol{I}_n$ is positive definite diagonal matrix due to the upper bound on $\epsilon$. Define $\boldsymbol{C}^\epsilon$ as the Cholesky factor of $\boldsymbol{D} + \tilde{\boldsymbol{S}}^\epsilon - \epsilon\boldsymbol{I}_n$.

Now we show that the covariance of observed vector under $\mathfrak{C}'$ is the same as that under $\mathfrak{C}$. Given the independence of the noise vector $\underline{Z}$ and the confounder vector $\underline{H}$, the covariance matrix of the observed vector $\boldsymbol{\Sigma}^\epsilon = E[\underline{X}^\epsilon(\underline{X}^\epsilon)^\top]$ equals

$$\begin{aligned}
\boldsymbol{\Sigma}^{\boldsymbol{\epsilon}} &= \boldsymbol{B}^\epsilon(\boldsymbol{B}^\epsilon)^\top + \boldsymbol{C}^\epsilon(\boldsymbol{C}^\epsilon)^\top \\
&= \boldsymbol{B}^\epsilon(\boldsymbol{B}^\epsilon)^\top + (\boldsymbol{D} + \tilde{\boldsymbol{S}}^\epsilon - \epsilon\boldsymbol{I}_n) \\
&= \boldsymbol{B}^\epsilon(\boldsymbol{B}^\epsilon)^\top + (\boldsymbol{D} + \boldsymbol{S} - \boldsymbol{B}^\epsilon(\boldsymbol{B}^\epsilon)^\top) \qquad (6) \\
&= \boldsymbol{\Sigma} \qquad (7)
\end{aligned}$$

where (6) follows from the definition of $\tilde{\boldsymbol{S}}^\epsilon$. As $\underline{X}$ and $\underline{X}^\epsilon$ are zero-mean Gaussian random variables, the equality of their covariance matrices implies that $\underline{X}^\epsilon \stackrel{d}{=} \underline{X}$. This completes the proof as there are infinitely many $\epsilon$ that meets the aforementioned two conditions.

(necessity) If the observed vector $\underline{X}$ in the SCM $\mathfrak{C}(\boldsymbol{A}, \boldsymbol{B}, \boldsymbol{C})$ is independent, then the covariance matrix $\boldsymbol{\Sigma}$ of $\underline{X}$ is diagonal. This implies that $\boldsymbol{A}$ is a zero matrix. Similarly, Assumption 1 implies that $\boldsymbol{B}$ is a zero matrix. Hence, $\boldsymbol{C}$ can be uniquely identified as the Cholesky factor of $\boldsymbol{\Sigma}$. $\square$

**Proof of Proposition 1.** Denote SCM $\mathfrak{C}$ by $\mathfrak{C}(\boldsymbol{M}, \boldsymbol{Q})$. Consider confounder-free SCM $\mathfrak{C}'(\boldsymbol{0}, \boldsymbol{Q}')$ where $\boldsymbol{Q}'$ is the Cholesky factor of $\boldsymbol{M}\boldsymbol{M}^\top + \boldsymbol{Q}\boldsymbol{Q}^\top$, i.e.,

$$\boldsymbol{Q}'\boldsymbol{Q}'^\top = \boldsymbol{M}\boldsymbol{M}^\top + \boldsymbol{Q}\boldsymbol{Q}^\top. \tag{8}$$

On the other hand, According to (2) and Given the independence of noise and confounder case, the covariance matrix of the observed variables in $\mathfrak{C}$ equals $\boldsymbol{M}\boldsymbol{M}^\top + \boldsymbol{Q}\boldsymbol{Q}^\top$, and equals $\boldsymbol{Q}'\boldsymbol{Q}'^\top$ in $\mathfrak{C}'$. In view of (8), the two covariance matrices match, implying equivalence in distribution of $\underline{X}$ and $\underline{X}'$ as they are zero-mean Gaussian variables, completing the proof. A second proof is provided in the following, by considering a Gaussian SCM $\mathfrak{C}'$ in the form of (1) (rather than (2)) and obtaining $(\boldsymbol{A}', \boldsymbol{0}, \boldsymbol{C}')$ for $\mathfrak{C}'$. To prove the converse of proposition, we can follow the proof of the sufficiency of Theorem 1 by letting $\boldsymbol{M} = 0$.

$\square$

**Second proof of Proposition 1:** Consider SCM $\mathfrak{C}(\boldsymbol{A}, \boldsymbol{B}, \boldsymbol{C})$. Without loss of generality assume that matrix $\boldsymbol{A}$ is lower triangular as this can be obtained by re-ordering the observed variables. we prove the proposition for $\boldsymbol{B} \neq \boldsymbol{0}$. Consider the following Gaussian SCM, denoted by $\mathfrak{C}'(\boldsymbol{A}', \boldsymbol{0}, \boldsymbol{C}')$, without confounder variables:

$$\underline{X}' := \boldsymbol{A}'\underline{X}' + \boldsymbol{C}'\underline{Z}',$$

where $\boldsymbol{A}'$ is a lower triangular matrix with zero diagonal entries whose entries below the main diagonal in the $(i+1)^{\text{th}}$ row equals $\underline{A}'^{i+1} = \underline{\Sigma}^{i+1}(\boldsymbol{\Sigma}^{ii})^{-1}$ for $i \geq 1$, where

$$\boldsymbol{\Sigma}^{ii} = E[[X_1, \dots, X_i]^\top [X_1, \dots, X_i]], \qquad \underline{\Sigma}^{i+1} = E[X_{i+1}[X_1, \dots, X_i]]. \tag{9}$$

Due to the dependency among the observed variables, $E[\underline{X}\underline{X}^\top]$ is not diagonal. Consequently, the matrix $\boldsymbol{A}'$ is non-zero. Also, let $\boldsymbol{C}'$ be a diagonal matrix whose diagonal entries are defined as follows for $i \geq 1$:

$$C'_{i+1,i+1} = E[X_{i+1}^2] - (\underline{\Sigma}^{i+1})(\boldsymbol{\Sigma}^{ii})^{-1}(\underline{\Sigma}^{i+1})^\top, \tag{10}$$

and $C'_{11} = E(X_1^2)$. Due to the Cauchy inequality (Tripathi, 1999), $C'_{i+1,i+1}$ is positive, and therefore, SCM $\boldsymbol{C}'$ is well-defined. Let $\boldsymbol{\Sigma}'$ denote the covariance matrix of $\underline{X}'$ under SCM $\mathfrak{C}'$. We show by induction on $i$ that $\boldsymbol{\Sigma}'^{nn} = \boldsymbol{\Sigma}^{nn}$. For $i = 1$, $\boldsymbol{\Sigma}^{11} = \boldsymbol{\Sigma}'^{11}$. For every $j \leq i$, we assume $\boldsymbol{\Sigma}^{jj} = \boldsymbol{\Sigma}'^{jj}$. Then, for $i+1$, it holds that

$$\begin{aligned}
\underline{\Sigma}'^{i+1} &= E[X'_{i+1}[X'_1, \dots, X'_i]] \\
&= E[\underline{A}'^{i+1}[X'_1, \dots, X'_i]^\top [X'_1, \dots, X'_i]] + C_{i+1,i+1} E[Z'_{i+1}[X'_1, \dots, X'_i]] \\
&\overset{(a)}{=} \underline{A}'^{i+1}\boldsymbol{\Sigma}'^{ii} + 0 \\
&\overset{(b)}{=} \underline{\Sigma}^{i+1}(\boldsymbol{\Sigma}^{ii})^{-1}\boldsymbol{\Sigma}^{ii} \tag{11} \\
&= \underline{\Sigma}^{i+1}, \tag{12}
\end{aligned}$$

where $(a)$ holds due to the fact that $X'_1, \dots, X'_i$ depend only on $Z'_1, \dots, Z'_i$ which are independent of $Z'_{i+1}$ by definition and therefore, $E\left[Z'_{i+1}[X'_1, \dots, X'_i]\right] = 0$, Also, $(b)$ is due to the induction assumption. Therefore, $\underline{\Sigma}'^{i+1} = \underline{\Sigma}^{i+1}$. Now, we prove that $E[X'^2_{i+1}] = E[X_{i+1}^2]$ for $i = 1, \dots, n$:

$$\begin{aligned}
E[X'^2_{i+1}] &= E\left[(\underline{A}'^{i+1}[X'_1, \dots, X'_i]^\top + C'_{i+1,i+1}Z'_{i+1})^2\right] \\
&= \underline{A}'^{i+1}\boldsymbol{\Sigma}'^{ii}\underline{A}'^{i+1^\top} + E[C'^2_{i+1,i+1}] \\
&= (\underline{\Sigma}^{i+1})(\boldsymbol{\Sigma}^{ii})^{-1}(\underline{\Sigma}^{i+1})^\top + E[X_{i+1}^2] - (\underline{\Sigma}^{i+1})(\boldsymbol{\Sigma}^{ii})^{-1}(\underline{\Sigma}^{i+1})^\top \\
&= E[X_{i+1}^2]. \tag{13}
\end{aligned}$$

Thus, (12) and (13) imply $\boldsymbol{\Sigma}'^{i+1,i+1} = \boldsymbol{\Sigma}^{i+1,i+1}$, which completes the induction step. Therefore, $\boldsymbol{\Sigma}'^{nn} = \boldsymbol{\Sigma}^{nn}$. On the other hand, Gaussian variables $\underline{X}$ and $\underline{X}'$ are defined as zero mean. Hence,

the mean and covariance matrices of the observed variables under SCM $\mathfrak{C}'$ match those under $\mathfrak{C}$, implying $\underline{X}' \overset{d}{=} \underline{X}$. $\qquad\square$

**Proof of Proposition 2:** In the proof of Theorem 1, under Assumption 1, we showed the existence of a vector $\underline{H} \in \mathbb{R}^m$, where $m$ represents the number of columns in $\mathcal{L}^\epsilon$ with more than one non-zero entry. Matrix $\mathcal{L}^\epsilon$ was derived from the Cholesky decomposition. On the other hand, being lower triangular, every Cholesky factor admits at most $n-1$ columns with more than one non-zero entry. Consequently, all SCMs considered in this proof satisfy $m < n$, indicating that the SCM is not identifiable even when the number of confounders is less than the number of observed variables.

Now, if Assumption 1 is violated, there exists a confounder-free SCM equivalent to SCM $\mathfrak{C}$. By decomposing $n-1$ Gaussian noise of this SCM into two subcomponents, one for the confounder, one for the noise variable – which can happen by infinitely different ways– a new equivalent SCM with $n-1$ Gaussian confounders is generated. $\square$

For the proof of Theorem 2, we need the following lemmas.

**Lemma 5 (based on (Pavan & Miranda, 2018), Theorem 3)** *Let $U_1, U_2, \ldots, U_n$ be real and mutually independent random variables. Define $V_1$ and $V_2$ as two real random variables, each equal to a linear combination of $U_1, U_2, \ldots, U_n$ in distribution sense,*

$$\begin{cases} V_1 \overset{d}{=} a_1 U_1 + \ldots + a_n U_n, \\ V_2 \overset{d}{=} b_1 U_1 + \ldots + b_n U_n, \end{cases} \tag{14}$$

*where $a_i$ and $b_i$ for $i = 1, \ldots, n$ are real constants. If $V_1$ and $V_2$ are independent, then for each index $i$ such that $a_i b_i \neq 0$, it follows that $U_i$ must be either a constant or a Gaussian random variable.*

**Proof of Lemma 5:** This lemma is similar to Theorem 3 in (Pavan & Miranda, 2018) with the key difference that the theorem assumed equality rather than equality in distribution sense. There, using the equality, $\Phi_{V_1, V_2}(t_1, t_2) = \prod_{i=1}^{n} \Phi_{U_i}(t_1 a_i + t_2 b_i)$ is concluded in the beginning of the proof, where $\Phi_{V_1, V_2}(t_1, t_2)$ is the joint CF of $V_1$ and $V_2$, and $\Phi_{U_i}(t)$ is the CF of $U_i$. Here, we can conclude the same using the equality in distribution and the independence of $U_i$'s. Therefore, the rest of the proof of this lemma follows the same arguments as those in Theorem 3 in (Pavan & Miranda, 2018). $\square$

**Lemma 6** *Let $\underline{U}$ be a vector of $m$ mutually independent random variables, none of which are Gaussian or constant, with covariance matrix $\boldsymbol{I}_m$. Let $\underline{V}$ be a vector of $n$ mutually independent random variables with covariance matrix $\boldsymbol{I}_n$. If $\underline{V} \overset{d}{=} \boldsymbol{R}\underline{U}$, then $\boldsymbol{R} \in \mathbb{R}^{n \times m}$ has orthogonal rows, and each column of $\boldsymbol{R}$ has at most one non-zero entry.*

**Proof of Lemma 6:** Given $\underline{V} \overset{d}{=} \boldsymbol{R}\underline{U}$, we have $E[\underline{V}\underline{V}^\top] = \boldsymbol{R}E[\underline{U}\underline{U}^\top]\boldsymbol{R}^\top$. Since $\underline{V}$ and $\underline{U}$ have identity covariance matrices, $\boldsymbol{R}\boldsymbol{R}^\top = \boldsymbol{I}_n$. Thus, $\boldsymbol{R}$ has orthogonal rows. Now, to show that each column of $\boldsymbol{R}$ has at most one non-zero entry, we proceed by contradiction. Suppose that there exists a column of $\boldsymbol{R}$, say the $j^{\text{th}}$, with at least two non-zero entries. Then $U_j$ appears in the linear combinations of at least two independent random variables in $\underline{V}$. This contradicts Lemma 5, as then $U_j$ must be Gaussian or constant. $\qquad\square$

**Lemma 7** *Let $\boldsymbol{\Sigma} \in \mathbb{R}^{n \times n}$ be a positive definite matrix. There are at most $n!$ different matrices $\boldsymbol{Q}$ such that $\boldsymbol{\Sigma} = \boldsymbol{Q}\boldsymbol{Q}^\top$ where $Q_{ij}Q_{ji} = 0$ for every $i, j \in \{1, 2, \ldots, n\}$, $i \neq j$. For every matrix $\boldsymbol{Q}$, there exists a permutation matrix $\boldsymbol{P}$ such that $\boldsymbol{Q} = \boldsymbol{P}^\top \boldsymbol{Q_p} \boldsymbol{P}$ where $\boldsymbol{Q_p}$ is the lower triangular matrix obtained by the Cholesky decomposition of $\boldsymbol{P}\boldsymbol{\Sigma}\boldsymbol{P}^\top$.*

**Proof of Lemma 7**: Due to the structure of $\boldsymbol{Q}$, there is a permutation matrix $\boldsymbol{P}$ that reshapes $\boldsymbol{Q}$ as a lower triangular matrix denoted by $\boldsymbol{Q_p}$, i.e., $\boldsymbol{Q_p} = \boldsymbol{P}\boldsymbol{Q}\boldsymbol{P}^\top$. Due to the fact that $\boldsymbol{P}^\top \boldsymbol{P} = \boldsymbol{I_n}$, we have $\boldsymbol{Q} = \boldsymbol{P}^\top \boldsymbol{Q_p}\boldsymbol{P}$, which in view of $\boldsymbol{\Sigma} = \boldsymbol{Q}\boldsymbol{Q}^\top$ yields:

$$\boldsymbol{\Sigma} = \boldsymbol{Q}\boldsymbol{Q}^\top$$
$$\Rightarrow \boldsymbol{\Sigma} = \boldsymbol{P}^\top \boldsymbol{Q_p}\boldsymbol{Q_p}^\top \boldsymbol{P}$$
$$\Rightarrow \boldsymbol{P}\boldsymbol{\Sigma}\boldsymbol{P}^\top = \boldsymbol{Q_p}\boldsymbol{Q_p}^\top$$

The last equality is the Cholesky decomposition of $\boldsymbol{P\Sigma P}^\top$ since $\boldsymbol{Q_p}$ is a lower triangular matrix. The Cholesky decomposition for real-valued positive-definite matrices is unique (Golub & Van Loan, 1996). Therefore, corresponding to every permutation matrix $\boldsymbol{P}$, there is a unique Cholesky factor of $\boldsymbol{P\Sigma P}^\top$ or equivalently a unique $\boldsymbol{Q_p} = \boldsymbol{PQP}^\top$. Given that there are $n!$ permutation matrices $\boldsymbol{P}$, there are $n!$ matrices $\boldsymbol{Q} = \boldsymbol{P}^\top \boldsymbol{Q_p} \boldsymbol{P}$ satisfying $\boldsymbol{\Sigma} = \boldsymbol{QQ}^\top$, although some may be identical. Hence, there are at most $n!$ distinct matrices $\boldsymbol{Q}$ satisfying $\boldsymbol{\Sigma} = \boldsymbol{QQ}^\top$. $\qquad\square$

**Proof of Theorem 2:** Consider the following two SCMs $\mathfrak{C}$ and $\mathfrak{C}'$ where $\underline{X} \stackrel{d}{=} \underline{X}'$.

$$\mathfrak{C}: \underline{X} = \boldsymbol{M}\underline{H} + \boldsymbol{Q}\underline{Z}, \qquad\qquad \mathfrak{C}': \underline{X}' = \boldsymbol{M}'\underline{H}' + \boldsymbol{Q}'\underline{Z}'. \tag{15}$$

Without loss of generality, we consider the causal structure order $X_1 \succ X_2 \succ \ldots X_n$ for SCM $\mathfrak{C}$. Therefore, $\boldsymbol{Q}$ is a lower triangular matrix. The proof consists of two steps. First, we show that $\boldsymbol{M} = \boldsymbol{M}'$. Second, we show the existence of at most $n!$ noise matrices $\boldsymbol{Q}'$.

**Step 1.** Due to Assumption 4, the rank of matrix $\boldsymbol{M}$ is $k$. Therefore, for an arbitrary $m^{\text{th}}$ row of $\boldsymbol{M}$, there are $k-1$ other rows that together with row $m$, form a full-rank $k \times k$ sub-matrix denoted $\tilde{\boldsymbol{M}}$. Denote the vector of variables corresponding to these rows by $\underline{\tilde{X}}$, which form an informative variable set. The equality $\mathbb{P}(\underline{X}) = \mathbb{P}(\underline{X}')$ implies $\phi_{\underline{X}}(\underline{t}) = \phi_{\underline{X}'}(\underline{t})$, where $\underline{t} = [t_1, t_2, \ldots, t_n]^\top$, because two random variables have the same CF if and only if they have the same PDF. Now, considering Remark 6 and the independence of noise and confounders, we have:

$$\phi_{\boldsymbol{M}\underline{H}}(\underline{t})\phi_{\boldsymbol{Q}\underline{Z}}(\underline{t}) = \phi_{\boldsymbol{M}'\underline{H}'}(\underline{t})\phi_{\boldsymbol{Q}'\underline{Z}'}(\underline{t}). \tag{16}$$

Due to Assumption 2, the non-Gaussian (resp. Gaussian) components on both sides must match. Thus,

$$\phi_{\boldsymbol{M}\underline{H}}(\underline{t}) = \phi_{\boldsymbol{M}'\underline{H}'}(\underline{t}). \tag{17}$$

Constraining this equation to the informative observed variables $\underline{\tilde{X}}$ yields $\phi_{\tilde{\boldsymbol{M}}\underline{H}}(\underline{t}) = \phi_{\tilde{\boldsymbol{M}}'\underline{H}'}(\underline{t})$. This is equivalent to $\phi_{\underline{H}}(\tilde{\boldsymbol{M}}^\top \underline{t}) = \phi_{\underline{H}'}(\tilde{\boldsymbol{M}}'^\top \underline{t})$, due to Lemma 2. For an arbitrary value of $\underline{w} \in \mathbb{R}^k$, set $\underline{t} = (\tilde{\boldsymbol{M}}^\top)^{-1}\underline{w}$. Therefore,

$$\begin{aligned}
&\phi_{\underline{H}}(\tilde{\boldsymbol{M}}^\top (\tilde{\boldsymbol{M}}^\top)^{-1}\underline{w}) = \phi_{\underline{H}'}(\tilde{\boldsymbol{M}}'^\top (\tilde{\boldsymbol{M}}^\top)^{-1}\underline{w}) \\
\Rightarrow &\phi_{\underline{H}}(\underline{w}) = \phi_{\underline{H}'}(\tilde{\boldsymbol{M}}'^\top (\tilde{\boldsymbol{M}}^\top)^{-1}\underline{w}) \\
\Rightarrow &\underline{H} \stackrel{d}{=} \tilde{\boldsymbol{M}}^{-1}\tilde{\boldsymbol{M}}'\underline{H}'.
\end{aligned} \tag{18}$$

Define $\boldsymbol{R} \in \mathbb{R}^{k \times k'}$ as $\boldsymbol{R} = \tilde{\boldsymbol{M}}^{-1}\tilde{\boldsymbol{M}}'$. In view of (1) and Assumption 2, the conditions of Lemma 6 are met for $\underline{V} = \underline{H}$ and $\underline{U} = \underline{H}'$. Thus, $\boldsymbol{R}$ is full row rank, implying that $k' \geq k$. Now we show that $k' = k$. According to the definition of $\boldsymbol{R}$, $\tilde{\boldsymbol{M}}' = \tilde{\boldsymbol{M}}\boldsymbol{R}$, which in view of (18) results in $\Phi_{\underline{H}}(\underline{w}) = \Phi_{\underline{H}'}(\boldsymbol{R}\underline{w})$. Thus, $\boldsymbol{R}$ is independent of row $m$, resulting in $\boldsymbol{M}' = \boldsymbol{M}\boldsymbol{R}$ as row $m$ was arbitrary. Now, if $k' > k$, the columns of $\boldsymbol{R}$ are linearly dependent. Since $\boldsymbol{M}' = \boldsymbol{M}\boldsymbol{R}$, the columns of $\boldsymbol{M}'$ are also linearly dependent, which contradicts Assumption 4. Thus, $k' = k$. Hence, $\boldsymbol{R}$ is a permutation matrix as it is full rank with orthogonal rows and each of its columns have at most one none-zero entry. On the other hand, Assumption 3 implies that both $\boldsymbol{M}$ and $\boldsymbol{M}$ are lexicographically sorted. Hence, $\boldsymbol{R}$ is the identity matrix and $\boldsymbol{M}' = \boldsymbol{M}$.

**Step 2.** In view of (16), Assumption 2 implies

$$\phi_{\boldsymbol{Q}\underline{Z}}(\underline{t}) = \phi_{\boldsymbol{Q}'\underline{Z}'}(\underline{t}) \tag{19}$$

where $\underline{Z}$ and $\underline{Z}'$ are zero-mean Gaussian random vectors of the same dimension $n$. Therefore, (19) is equivalent to the covariance matrices $\boldsymbol{\Sigma} = \boldsymbol{QQ}^\top$ and $\boldsymbol{\Sigma}' = \boldsymbol{Q}'\boldsymbol{Q}'^\top$ being equal, i.e., $\boldsymbol{\Sigma} = \boldsymbol{\Sigma}'$. Now, it follows from the definition of SCMs (see (1) and (2)) that $\boldsymbol{\Sigma}$ and $\boldsymbol{\Sigma}'$ are positive definite matrices. Moreover, $\boldsymbol{Q}$ as well as $\boldsymbol{Q}'$ satisfy $Q_{ij}Q_{ji} = 0$ and $Q'_{ij}Q'_{ji} = 0$ for every $i, j \in \{1, 2, \ldots, n\}, i \neq j$. Therefore, according to Lemma 7, there are at most $n!$ possible $\boldsymbol{Q}'$ matrices.

Now, we prove the third part of the Theorem. Substituting $\tilde{\boldsymbol{M}}^{-1}\tilde{\boldsymbol{M}}' = \boldsymbol{I}_k$ in (18), $\Phi_{\underline{H}}(\underline{w}) = \Phi_{\underline{H}'}(\underline{w})$. Additionally, (15) implies that

$$\tilde{\boldsymbol{M}}^{-1}\underline{\tilde{X}} = \underline{H} + \tilde{\boldsymbol{M}}^{-1}\tilde{\boldsymbol{Q}}\underline{Z}.$$

The proof is complete considering the independence of noise and confounders and according to Remark 6 and Lemma 2. $\qquad\square$

**Proof of Lemma 1:** The following equations hold:

$$\boldsymbol{Q}' = \boldsymbol{Q}$$
$$\Leftrightarrow \boldsymbol{P}^\top \boldsymbol{Q}_{\boldsymbol{P}} \boldsymbol{P} = \boldsymbol{Q} \tag{20}$$
$$\Leftrightarrow \boldsymbol{Q}_{\boldsymbol{P}} = \boldsymbol{P}\boldsymbol{Q}\boldsymbol{P}^\top \tag{21}$$
$$\Leftrightarrow \boldsymbol{P}\boldsymbol{Q}\boldsymbol{P}^\top \text{ is a lower triangular matrix} \tag{22}$$

where (20) holds due to Lemma 7, (21) is a result of $\boldsymbol{P}\boldsymbol{P}^\top = \boldsymbol{I}_n$, and (22) holds in the forward direction since $\boldsymbol{Q}_{\boldsymbol{P}}$ is the lower triangular matrix. Now, we prove the reverse direction of (22). It is clear that $(\boldsymbol{P}\boldsymbol{Q}\boldsymbol{P}^\top)(\boldsymbol{P}\boldsymbol{Q}\boldsymbol{P}^\top)^\top = \boldsymbol{P}\boldsymbol{\Sigma}\boldsymbol{P}^\top$. Therefore, if $\boldsymbol{P}\boldsymbol{Q}\boldsymbol{P}^\top$ is a lower triangular matrix, it is exactly the result of the Cholesky decomposition of $\boldsymbol{P}\boldsymbol{\Sigma}\boldsymbol{P}^\top$. According to Lemma 7, $\boldsymbol{Q}_{\boldsymbol{P}}$ is also derived from the Cholesky decomposition of $\boldsymbol{P}\boldsymbol{\Sigma}\boldsymbol{P}^\top$. Therefore, due to the uniqueness of Cholesky decomposition, $\boldsymbol{Q}_{\boldsymbol{P}} = \boldsymbol{P}\boldsymbol{Q}\boldsymbol{P}^\top$. $\qquad\square$

The following result is straightforward.

**Lemma 8** *Consider a vector of random variables $\underline{X} = [X_1, \ldots, X_n]^\top$. A Permutation matrix $\boldsymbol{P}$ has a one-to-one relationship with the permutation order $\mathcal{O}_{\boldsymbol{P}}(\underline{X})$, i.e, $\boldsymbol{P} \leftrightarrow \mathcal{O}_{\boldsymbol{P}}(\underline{X})$, such that $P_{ij} = 1$ if and only if $X_j$ is located at the $i^{th}$ row in $\boldsymbol{P}\underline{X}$.*

**Proof of Theorem 3:** (PART 1): $\mathfrak{C}$ and $\mathfrak{C}'$ are distinct SCMs if and only if

$$\boldsymbol{P}\boldsymbol{Q}\boldsymbol{P}^\top \text{ is not a lower triangular matrix} \tag{23}$$
$$\Leftrightarrow \exists i, j \in \{1, 2, \ldots, n\}; i < j, (\boldsymbol{P}\boldsymbol{Q}\boldsymbol{P}^\top)_{ij} \neq 0 \tag{24}$$
$$\Leftrightarrow \exists i, j, k, l \in \{1, 2, \ldots, n\}; P_{ik} = 1, P_{jl} = 1, i < j, Q_{kl} \neq 0 \tag{25}$$
$$\Leftrightarrow O_P \notin \mathfrak{D}_{\mathfrak{C}} \tag{26}$$

where (23) follows from Lemma 1 and (24) follows from the definition of a lower triangular matrix. The properties of the permutation matrix lead to (25) where in the $i^{th}$ row of $\boldsymbol{P}$, only one column $k$ is non-zero and equal to one. To show that (26) holds, note that according to Lemma 8, $P_{ik} = 1$ and $P_{jl} = 1$ imply that $X'_k$ and $X'_l$ are ranked $i^{th}$ and $j^{th}$, respectively, in the causality order $\mathcal{O}_p$. On the other hand, since $i < j$, $X'_k$ precedes $X'_l$ in $\mathcal{O}_P$, i.e., $X'_k \succ X'_l$. Furthermore, $Q_{kl} \neq 0$ indicates that noise variable $Z_l$ appears in the equation of $X_k$ in (2) for SCM $\mathfrak{C}$. This implies the causality order $X_l \succ X_k$ in every causality order in $\mathfrak{D}_{\mathfrak{C}}$. Therefore, $\mathcal{O}_p \notin \mathfrak{D}_{\mathfrak{C}}$.

(Part 2) Multiply $\boldsymbol{P}$ in both sides of (2) to obtain

$$\boldsymbol{P}\underline{X} = \boldsymbol{P}\boldsymbol{M}\underline{H} + (\boldsymbol{P}\boldsymbol{Q}\boldsymbol{P}^\top)(\boldsymbol{P}\underline{Z}). \tag{27}$$

Here, $\boldsymbol{P}\boldsymbol{Q}\boldsymbol{P}^\top$ is not necessarily a lower triangular matrix. Now since SCM $\mathfrak{C}'$ shares the same $\boldsymbol{M}$ and only differs in $\boldsymbol{Q}'$, it holds that

$$\boldsymbol{P}\underline{X}' = \boldsymbol{P}\boldsymbol{M}\underline{H}' + (\boldsymbol{P}\boldsymbol{Q}'\boldsymbol{P}^\top)(\boldsymbol{P}\underline{Z}'). \tag{28}$$

Also, from Theorem 2, $\boldsymbol{Q}' = \boldsymbol{P}^\top \boldsymbol{Q}_p \boldsymbol{P}$, yielding

$$\boldsymbol{P}\underline{X}' = \boldsymbol{P}\boldsymbol{M}\underline{H}' + \boldsymbol{Q}_p(\boldsymbol{P}\underline{Z}'). \tag{29}$$

Since $\boldsymbol{Q}_p$ is lower triangular, the order defined by $\boldsymbol{P}\underline{X}'$ is a valid causality order of $\mathfrak{C}'$. On the other hand, the order defined by $\boldsymbol{P}\underline{X}'$ is the same as the permutation order $\mathcal{O}_{\boldsymbol{P}}(\underline{X})$, completing the proof. $\square$

**Proof of Corollary 1:** The causality order determines the permutation matrix $\boldsymbol{P}$ uniquely. Consequently, $\mathfrak{C}' = \mathcal{F}_{\mathfrak{C}}(\boldsymbol{P})$ is determined uniquely by the permutation matrix $\boldsymbol{P}$. $\qquad\square$

**Proof of Corollary 2:** In view of Theorem 2, Part 2, every SCM in the equivalence class of $\mathfrak{C}(\boldsymbol{M}, \boldsymbol{Q})$ can be written as $\mathfrak{C}'(\boldsymbol{M}, \boldsymbol{P}^\top \boldsymbol{Q}_p \boldsymbol{P})$ with $\boldsymbol{Q}_p$ being the lower triangular matrix obtained from the Cholesky decomposition of $\boldsymbol{P}\boldsymbol{Q}\boldsymbol{Q}^\top \boldsymbol{P}^\top$, for some permutation matrix $\boldsymbol{P}$. By Theorem 3, the causal order set of each SCM $\mathfrak{C}'(\boldsymbol{M}, \boldsymbol{P}^\top \boldsymbol{Q}_p \boldsymbol{P})$ includes the permutation order $\mathcal{O}_{\boldsymbol{P}}$. On the

other hand, SCM $\mathfrak{C}'$ respects the specified partial causal order. Consequently, a valid permutation matrix $P$ rearranges the rows of the observed variables $\mathcal{X}_i$ only within $\mathcal{X}_i$ itself. The number of such permutations, and in turn the size of the equivalence class, is given by $\prod_{i=1}^{m} |\mathcal{X}_i|!$. $\qquad\square$

**Proof of Corollary 3:** (sufficiency) Since all observed variables are isolated, matrix $Q$ is diagonal. Therefore, for any permutation matrix $P$, matrix $PQP^{\top}$ is diagonal, and hence, also lower triangular. Consequently, for any $P$, SCMs $\mathfrak{C}$ and $\mathfrak{C}'$ are the same (i.e., $Q = Q'$) according to Lemma 1, where $\mathcal{F}_{\mathfrak{C}}(P) = \mathfrak{C}'$. Thus, $\mathfrak{C}$ is uniquely identifiable. (necessity) Unique identifiability implies that for every permutation matrix $P$, $Q = Q'$ or equivalently $PQP^{\top}$, is lower triangular. This implies that $Q$ must be a diagonal matrix. In other words, all observed variables in $\mathfrak{C}$ must be isolated. $\qquad\square$

**Proof of Corollary 4:** First, we prove that an isolated set in SCM $\mathfrak{C}$ is also an isolated set in all other SCMs in the equivalence class of $\mathfrak{C}$. Without loss of generality, let $\mathcal{X} = \{X_1, \ldots, X_r\}$ be an isolated variable set for $\mathfrak{C}$. In view of Theorem 2, Part 2, every SCM in the equivalence class of $\mathfrak{C}(M, Q)$ can be written as $\mathfrak{C}'(M, P^{\top} Q_p P)$ with $Q_p$ being the lower triangular matrix obtained from the Cholesky decomposition of $PQQ^{\top}P^{\top}$, for some permutation matrix $P$. Now, consider the case where $P$ takes the block-diagonal form $\mathrm{Diag}(P_1, P_2)$ where $P_1$ is a $r \times r$ matrix. According to Definition 11, matrix $Q$ is block diagonal, and we have $\Sigma = QQ^{\top} = \mathrm{Diag}(\Sigma_1, \Sigma_2)$ as a block-diagonal matrix, where $\Sigma_1$ and $\Sigma_2$ represent the blocks corresponding to the isolated set and the remaining variables, respectively. Clearly, $P\Sigma P^{\top}$ remains block diagonal matrix. This implies that the Cholesky decomposition factor $Q_p$ and $Q' = P^{\top} Q P$ are also block diagonal. Therefore, the variable set $\mathcal{X}$ is isolated in every SCM $\mathfrak{C}'$. Let $\mathcal{C}$ be the set of all such SCMs $\mathfrak{C}'$ whose permutation matrix $P$ are block-diagonal. Now, consider the case where $P$ does not take the aforementioned block-diagonal form. Then it can be shown that there exists an SCM $\mathfrak{C}' \in \mathcal{C}$ whose causality order set includes the permutation order $\mathcal{O}_P$. Thus, according to Theorem 3, no new SCM arises from this permutation. Hence, $\mathcal{C}$ is the equivalence class of $\mathfrak{C}$. It follows that if $\mathfrak{C}$ consists of several isolated variable sets $\mathcal{X}_1, \ldots, \mathcal{X}_m$, all of which will be isolated in every SCM of the equivalence class. Moreover, each of these SCMs correspond to a permutation matrix $P$ in the block diagonal form $\mathrm{Diag}(P_1, \ldots, P_m)$ where $P_i$ is a $|\mathcal{X}_i| \times |\mathcal{X}_i|$ permutation matrix, for all $i = 1, \ldots, m$. On the other hand, there are at most $s!$ permutation matrices of dimension $s \times s$. This completes the proof. $\qquad\square$

**Proof of Remark 3-Part 3:** According to the proof of Theorem 1, when the observed vector is dependent, there exist infinitely many SCMs $(M^{\epsilon}, Q^{\epsilon})$ whose observed variables are equal in distribution with those of SCM $\mathfrak{C}$. Since $Q^{\epsilon}$ is a lower triangular matrix, the causality orders of all these SCMs are identical. $\qquad\square$

**Proof of Remark 4-Part 3:** We provide two SCMs $\mathfrak{C}'(0, Q')$ and $\mathfrak{C}''(0, Q'')$ that are in the equivalence class of SCM $\mathfrak{C}$ but have different causal orders. We choose $Q'$ as the Choleskey factor of $\Sigma = QQ^{\top}$. Hence, $Q'$ is a lower triangular but not diagonal matrix, indicating dependence among the observed variables. Consequently, there exists a permutation matrix $P$ such that $PQ'P^{\top}$ is not a lower triangular matrix. Define $Q'' = P^{\top} Q_p P$ where $Q_p$ is the lower triangular matrix obtained by the Cholesky decomposition of $P\Sigma P^{\top}$ (see according to Lemma 7). Now, in view of Lemma 7, $\Sigma = Q'' Q''^{\top}$, which according to $\Sigma = Q' Q'^{\top}$ implies that $\mathfrak{C}'$ and $\mathfrak{C}''$ are in the same equivalence class as SCM $\mathfrak{C}$. On the other hand, based on the proof of Lemma 1, $Q'' \neq Q'$ as $PQ'P^{\top}$ is not a lower triangular matrix. Then, similar to the proof of Theorem 3, it can be shown that $\mathfrak{C}'$ and $\mathfrak{C}''$ have different causality orders. $\qquad\square$

