# OpenReview forum: "Linear SCM Identification in the Presence of Confounders and Gaussian Noise"
_ICLR.cc/2025/Conference — ICLR 2025 Spotlight_

### Official Review · Reviewer_huy7 · 2024-11-02

**Soundness:** 4
**Presentation:** 3
**Contribution:** 3
**Rating:** 8
**Confidence:** 3

**Summary:**

The paper investigates the identifiability of noisy linear structural causal models (SCMs) in settings where confounders and noise variables have distributions with Gaussian and/or non-Gaussian components. Specifically, the authors investigate necessary and sufficient conditions for the SCM to be uniquely, finitely or infinitely identifiable, depending on the distributions of confounders and noise and certain assumptions on the causal structure of the model. Theorem 1 extends the known counter examples addressing identifiablity under Gaussian confounders.

Theorem 2 provides sufficiency conditions for linear SCM's to be finitely identifiable and provides a wonderful characterization in terms of the number of permutations and the characteristic function of the confounders. Theorem 3 characterizes when a permutation of an SCM leads to a new SCM and affects the causal order. Section 6 provides numerous examples and illustrations of the results.

**Strengths:**

Solid contributions: The paper provides significant insights into SCM identifiability, that extend existing work.

Clear writing and proof structure: The writing is clear and concise, with proofs that are both rigorous and easy to follow. The presentation of theoretical results is structured and clear. I enjoyed reading the article.

**Weaknesses:**

I think the writing/accessibility of the article could be improved slightly:

Grammar and Punctuation: There are minor punctuation issues in the introduction, particularly with the use of commas. For example in line 41, line 42, or "It is near impossible to claim that all causes of a specified process are considered, and hence, the consideration of confounders is inevitable" in line 49 (I would suggest "It is nearly impossible to claim that all causes of a specified process are considered; hence, the consideration of confounders is inevitable." or two separate sentences).

Certain definitions can be assumed to be known by the reader, such as Definition 3 and Definition 7 (along with Remark 2 and Remark 3). I would suggest moving these to the appendix to streamline the main text.

The examples illustrating Remarks 5 and 6 could use some further steps to help the reading along. This could be done in the appendix.

Typos:
line 67: "base on"
200: \ref used instead of \eqref{}
387: 'is' illustrated by SCMs:

**Questions:**

Full identifiability of a causal model is typically a strong ask. Often, one only cares for a small part of the model to be identifiable (e.g. some treatment effect). Do the findings of the authors (in particular Theorem 2) give immediate insight in when part of the model is uniquely identifiable, even when the full model is not?

---

> ### Author Response · Authors · 2024-11-25
>
> Thank you for your constructive feedback.
>
> > Full identifiability of a causal model is typically a strong ask. Often, one only cares for a small part of the model to be identifiable (e.g. some treatment effect). Do the findings of the authors (in particular Theorem 2) give immediate insight in when part of the model is uniquely identifiable, even when the full model is not?
>
> Great question! In response, we added two new corollaries with their proofs in the appendix to illustrate the results in the case of partial identifiability:
>
> **Definition** A *partial causal order for SCM $\mathfrak{C}$* is an order
> $\mathcal{X}_1 \succ \ldots \succ \mathcal{X}_m$, where $\mathcal{X}_1, \ldots, \mathcal{X}_m$ partition the observed variables $\{X_1, \ldots, X_n\}$. There is no path from any node in $\mathcal{X}_i$ to a node in $\mathcal{X}_j$ in the induced DAG for every $i, j \in \{1, 2, \ldots, m\}$ with $i > j$.
>
>
> **Corollary** Consider SCM $\mathfrak{C}(\mathbf{M}, \mathbf{Q})$ defined by (2), satisfying Assumptions 2 to 4, with $k$ confounders, $1 \leq k \leq n$. If a partial causal order $\mathcal{X}_1 \succ \ldots \succ \mathcal{X}_m$ is specified, then the size of the equivalence class of $\mathfrak{C}$ is at most $\prod_1^m |\mathcal{X}_i|!$.
>
> **Definition**  Given SCM (1),
>     an observed variable set {$X_{i_1},\ldots,X_{i_m}$} is said to be isolated if $A_{ab}=A_{ba}=0$ for all $a\in${$i_1,\ldots,i_m$} and $b\not\in${$i_1,\ldots,i_m$}.
>
> **Corollary** Consider SCM $\mathfrak{C}(\mathbf{M}, \mathbf{Q})$ defined by (2), satisfying Assumptions 2 to 4, with $k$ confounders, $1 \leq k \leq n$.
>      If $\mathfrak{C}$ consists of isolated variable sets $\mathcal{X}_1,\ldots,\mathcal{X}_m$, then so is every other SCM in the equivalence class of $\mathfrak{C}$.
>      Moreover, the size of the equivalence class is at most $\prod_1^m|\mathcal{X}_i|!$.

---

> ### Author Response · Authors · 2024-11-25
>
> Thank you for the points regarding the writing/accessibility. We fixed the mentioned typos and further improved the readability of the document. We also moved the stated definitions and remarks to the appendix.

---

### Official Review · Reviewer_AGFs · 2024-11-03

**Soundness:** 3
**Presentation:** 3
**Contribution:** 2
**Rating:** 6
**Confidence:** 3

**Summary:**

The paper discusses identifiability criteria for linear structural causal models (SCMs) with different scenarios for Gaussianity of confounders and noise, and varying assumptions. More specifically, the paper fills in the gap of identifiability for Gaussian noise and non-Gaussian confounders. The main proof technique involves analysis of characteristic functions and reliance on Gaussian properties. Some numerical examples are provided.

**Strengths:**

The paper shows that linear SCMs with Gaussian noise are finitely identifiable under the assumption that confounders are non-constant and cannot be decomposed into random variables where any are Gaussian (Assumption 2). This result fills in gaps in identifiability where earlier results already identified finite identifiability of SCMs with non-Gaussian noise and confounders. From this perspective, the paper contributes original new results.

In terms of significance, it is clear that the main results contribute to the field of linear SCMs and is a relevant topic for the conference. While the framework may be simple, the main results are interesting.

Finally, it is worthwhile to point out that the paper is clearly polished. The authors provided an extensive discussion of existing literature, problem formulations, main results, and concerns.

**Weaknesses:**

- The linear model framework restricts the scope of the paper
- As the authors also pointed out, there are no discussions of algorithms to find an SCM
- Novelty in proof techniques are limited and appear to be a mixture of gaussian properties, characteristic function analysis, and reliance on Gaussianity assumptions

While the points above raise concerns of weaknesses on the applicability of the results, the paper's main theoretical results are still meaningful and interesting.

**Questions:**

The paper does a good job of addressing multiple aspects of the problem. The largest concern is the scope of the problem and whether it has sufficient significance. Some questions and suggestions include:

1. It may be helpful to highlight the main theoretical tools used in the proof (i.e., the use of characteristic functions) and connect these tools with other contexts beyond the SCM. While the technique may be novel within applications to SCM, it is not particularly novel in the broader statistical literature. Are there other areas in machine learning that can significantly benefit from the techniques?

2. Regarding causal orders -- the SCM is uniquely identifiable given the causal order, provided gaussian noise and non-gaussian confounders. What if an expert only gives a partial order, would it still be uniquely identifiable in some scenarios?

3. Are there any existing results in identifying the SCM that can benefit from results presented in this paper? How are the results of identifiability connected with classical results in estimating linear model parameters?

4. A minor typo: Line 125 "novel approached".

**Details Of Ethics Concerns:**

No ethics concerns were found.

---

> ### Author Response · Authors · 2024-11-25
>
> Thank you for your constructive feedback.
>
> > 2. Regarding causal orders -- the SCM is uniquely identifiable given the causal order, provided gaussian noise and non-gaussian confounders. What if an expert only gives a partial order, would it still be uniquely identifiable in some scenarios?
>
> Great question! In response, we added two new corollaries with their proofs in the appendix to illustrate the results in the case of partial identifiability:
>
> **Definition** A *partial causal order for SCM $\mathfrak{C}$* is an order
> $\mathcal{X}_1 \succ \ldots \succ \mathcal{X}_m$, where $\mathcal{X}_1, \ldots, \mathcal{X}_m$ partition the observed variables $\{X_1, \ldots, X_n\}$. There is no path from any node in $\mathcal{X}_i$ to a node in $\mathcal{X}_j$ in the induced DAG for every $i, j \in \{1, 2, \ldots, m\}$ with $i > j$.
>
>
> **Corollary** Consider SCM $\mathfrak{C}(\mathbf{M}, \mathbf{Q})$ defined by (2), satisfying Assumptions 2 to 4, with $k$ confounders, $1 \leq k \leq n$. If a partial causal order $\mathcal{X}_1 \succ \ldots \succ \mathcal{X}_m$ is specified, then the size of the equivalence class of $\mathfrak{C}$ is at most $\prod_1^m |\mathcal{X}_i|!$.
>
> **Definition**  Given SCM (1),
>     an observed variable set {$X_{i_1},\ldots,X_{i_m}$} is said to be isolated if $A_{ab}=A_{ba}=0$ for all $a\in${$i_1,\ldots,i_m$} and $b\not\in${$i_1,\ldots,i_m$}.
>
> **Corollary** Consider SCM $\mathfrak{C}(\mathbf{M}, \mathbf{Q})$ defined by (2), satisfying Assumptions 2 to 4, with $k$ confounders, $1 \leq k \leq n$.
>      If $\mathfrak{C}$ consists of isolated variable sets $\mathcal{X}_1,\ldots,\mathcal{X}_m$, then so is every other SCM in the equivalence class of $\mathfrak{C}$.
>      Moreover, the size of the equivalence class is at most $\prod_1^m|\mathcal{X}_i|!$.

---

> ### Author Response · Authors · 2024-11-25
>
> > A minor typo: Line 125 "novel approached".
>
> Thank you. It is now fixed.

---

> ### Author Response · Authors · 2024-11-26
>
> > Are there any existing results in identifying the SCM that can benefit from results presented in this paper? How are the results of identifiability connected with classical results in estimating linear model parameters?
>
> In Structural Causal Models (SCMs), the primary goal is to uncover the causal structure, including the causal order among observed variables and latent confounders. This involves modeling not just observed variables but also latent factors that may influence both observed and other latent variables.
>
> In contrast, system identification focuses on deriving a mathematical model (often linear or nonlinear) that explains input-output relationships in dynamic systems, without explicitly addressing causal structure. The emphasis is on fitting a model to observed data rather than inferring causality.
>
> Our results contribute to SCM identifiability by addressing causal structure recovery, which aligns more closely with SCM objectives than traditional system identification. While classical parameter estimation in linear models informs system identification, our work extends these principles to address causal identifiability, particularly in the presence of latent confounders. This bridge between SCMs and parameter estimation offers potential insights for advancing SCM methodologies.

---

> ### Author Response · Authors · 2024-11-26
>
> > It may be helpful to highlight the main theoretical tools used in the proof (i.e., the use of characteristic functions) and connect these tools with other contexts beyond the SCM. While the technique may be novel within applications to SCM, it is not particularly novel in the broader statistical literature.
>
> We employ characteristic functions as a key tool to establish identifiability results for linear structural causal models (SCMs). By utilizing characteristic functions, we can analytically separate the effects of Gaussian noise from non-Gaussian confounders in the observed variables. This separation is essential for deriving conditions under which an SCM is uniquely or finitely identifiable, especially in cases where traditional methods struggle due to the presence of confounding variables. We also linked the notion of causality order to permutation matrices, which allowed us to find the exact form of the other SCMs in the equivalence class, and in turn, determine the size of the class.
>
> Moreover, our results are not just an immediate consequence of using characteristic functions. We also integrate several linear algebra techniques and fundamental mathematical concepts throughout our proofs. These include, but are not limited to matrix decompositions, properties of positive definite matrices, rank conditions, and the careful handling of linear dependencies and independence among variables. By combining these mathematical tools with characteristic functions, we provide a rigorous framework that advances the understanding of identifiability in systems influenced by both observed and unobserved variables.
>
> This methodology connects to broader statistical literature, particularly in independent component analysis (ICA) and signal processing, where characteristic functions help decompose complex signals into independent components. By leveraging these functions, the paper integrates classical probability theory with modern causal inference techniques, offering a novel framework that enhances the understanding of identifiability in systems influenced by both observed and unobserved variables.
>
> > Are there other areas in machine learning that can significantly benefit from the techniques?
>
> The use of characteristic functions in this context has potential benefits for other areas in machine learning that deal with complex probability distributions and latent variables. For example, in generative models like variational autoencoders (VAEs) and normalizing flows, characteristic functions could improve the modeling of non-Gaussian latent spaces. In reinforcement learning and time series analysis, they might enhance the handling of stochastic processes and uncertainties, leading to more robust and reliable algorithms.

---

> ### Author Response · Authors · 2024-11-26
>
> >Weaknesses: The linear model framework restricts the scope of the paper.
>
> We acknowledge the concern that focusing on linear models might seem to limit the scope of our paper. However, we believe our work makes significant contributions within this framework. The identifiability of linear structural causal models (SCMs) with non-Gaussian confounders and Gaussian noise is a fundamental problem that presents nontrivial challenges. By addressing these challenges rigorously, we provide deeper insights into the identifiability of SCMs, which is essential for both theoretical advancement and practical applications.
>
> Moreover, many nonlinear systems can be linearized and approximated around their operating point as linear models. This makes our findings highly relevant, as they can be applied to a wide range of systems where linear approximations are valid locally. By thoroughly exploring the linear case, we lay a solid foundation that can inform and inspire future research into more complex, nonlinear models. Thus, our paper not only advances the understanding of identifiability in a critical and widely used class of models but also sets the stage for broader applications in causal inference and machine learning.

---

> > ### Comment · Reviewer_AGFs · 2024-11-27
> > **Thank you to authors for responses**
> >
> > Thank you for your responses. I have no further comments and am keeping the original score while leaning positively towards an accept, considering the concerns about scope are difficult to resolve but the results are interesting.

---

### Official Review · Reviewer_AfH3 · 2024-11-04

**Soundness:** 3
**Presentation:** 4
**Contribution:** 3
**Rating:** 8
**Confidence:** 4

**Summary:**

This paper is a theoretical work on the identifiability problem of the linear structure causal models. Specifically, the authors showed that under the general Gaussian assumptions for all components, the linear SCM is unidentifiable. The authors also provided mild sufficient conditions for the linear SCM mode to be (i) uniquely identifiable or (ii) finitely identifiable.

**Strengths:**

- Originality: The paper is a novel work that discusses the identifiability of the SCM model under different model assumptions.
- Quality: The paper provides a thorough discussion of the identifiability issue with complete theoretical justification. It is of high quality in terms of the results and writing.
- Clarity: The paper is organized in a mathematical paper style. It is easy to follow the assumptions and theorems.
- Significance: The result of this paper is appealing. The analysis is solid.

**Weaknesses:**

- One assumption seems confusing to me (see details in the questions part).
- Although the paper has solid theoretical results, it is a bit difficult to directly apply the results to any real application as the assumptions are hard to verify for real data.

**Questions:**

Major questions:
1. Assumption 2 requires "Confounders $H_i$ .... cannot be decomposed into multiple random variables where any is Gaussian".  This is difficult to verify. Can this condition be translated to any condition on the characteristic functions? For example, suppose we let $Z\sim N(0, v)$ and let another independent variable $W$ have the c.f. $\phi_W(t)=\phi_H(t)e^{vt^2}$. Then $\phi_W$ should not be in $L_2$ space for any $v>0$.
Also, this condition mentioned "multiple" random variables.
However, any density function can be approximated by an infinite mixture of Gaussians (in the weak topology sense). Does this limit scenario break this assumption?

2. Suppose for a real data where $\mathrm{X}$ and $\mathrm{H}$ are collected. How do we verify the model is identifiable?
3. For Definition 3, the rigorous definition for $X\stackrel{d}{=}Y$ should be: for any measurable set $A$ in the space, $P(X\in A)=P(Y\in B)$.  The equality in p.d.f. is a bit unrigorous.
4. In terms of finding the equivalence class in Sec. 6, is that equivalent to finding the invariant permutations of $\mathbf{\Sigma}$, i.e. {$\mathbf{P}\in\mathcal P: \mathbf{P}\mathbf{\Sigma}\mathbf{P}^T$}, where $\mathcal P$ is the permutation class? If so, one can divide the indices {1,2,.., n} (corresponding to rows/cols of $\mathbf{\Sigma}$) into equivalent classes ($i\sim j$ if and only $\Sigma_{ik}=\Sigma_{jk}\ \forall k$). The invariant permutation of $\mathbf{\Sigma}$ is group-isomorphic to the direct product of the permutation groups of the equivalent classes.

Minor issues:
1. For narrative citations, please remove the paratheses.
2. Pg. 2. Ln 56. Add a space between "," and "$\mathbf{B}$".
3. Remarks 4 and 5 are important results. They can be transformed into two Theorems/Propositions.

---

> ### Author Response · Authors · 2024-11-25
>
> Thank you for your constructive feedback.
>
> > In terms of finding the equivalence class in Sec. 6, is that equivalent to finding the invariant permutations of $\mathbf{\Sigma}$, i.e. {$\mathbf{P}\in\mathcal{P}: \mathbf{P\Sigma P}^\top$}, where  is the permutation class?
>
> No, the counter example is in Example 2: $\mathbf{P_4}$ and $\mathbf{P_1}$ correspond to the same SCM, but they result in different values of $\mathbf{P\Sigma P}^\top$. Intuitively, $\mathbf{P_4}$ and $\mathbf{P_1}$ result in different causal orders, but both belong to the causal order set of the original SCM. Thus, they do not result in distinct SCMs in view of Theorem 3.

---

> ### Author Response · Authors · 2024-11-25
>
> > Assumption 2 requires "Confounders $H_i$ ... cannot be decomposed into multiple random variables where any is Gaussian". This is difficult to verify. Can this condition be translated to any condition on the characteristic functions? For example, suppose we let $Z \sim N(0, \nu)$ and let another independent variable $W$ have the c.f. $\phi_W(t) = \phi_H(t)e^{\nu t^2}$. Then $\phi_W$ should not be in $L_2$ space for any $\nu > 0$.
>
> Indeed, this observation seems to be correct. The fastest decay of characteristic functions corresponds to the Gaussian distribution, which takes the form $ e^{-vt^2} $. Thus, if $ \phi_W(t) $ does not belong to $ L_2 $ for every $ v $, then this indicates that $ H $ does not have an exponential part. This can be useful in practice, if the range of valid variances $v$ can be specified.
>
> > Also, this condition mentioned "multiple" random variables. However, any density function can be approximated by an infinite mixture of Gaussians (in the weak topology sense). Does this limit scenario break this assumption?
>
> Thanks for pointing this out. We changed "multiple" to "finite".

---

> ### Author Response · Authors · 2024-11-25
>
> > For Definition 3, the rigorous definition for $X \overset{d}{=} Y$ should be: for any measurable set $A$ in the space, $P(X \in A) = P(Y \in B)$. The equality in p.d.f. is a bit unrigorous.
>
> Great point! We now used this definition:
>
> "Two real random variables  $X$ and $Y$ are equal in distribution sense denoted by $X\stackrel{d}{=}Y$ if
>     for any measurable set $\mathcal{A}$  in the common state space of $X$ and $Y$, $P(X\in\mathcal{A}) = P(Y\in\mathcal{A})$."

---

> ### Author Response · Authors · 2024-11-26
>
> > Suppose for a real data where $\mathbf{X}$ and $\mathbf{H}$ are collected. How do we verify the model is identifiable?
>
> To ensure our model's assumptions hold, we first verify that the data satisfies the necessary conditions—for example, confirming that $\mathbf{H}$ lacks a Gaussian component. This can be determined by analyzing its characteristic function for exponential decay [1]. Next, we estimate an initial structural causal model (SCM) $\mathfrak{C}$ from the data using algorithms like expectation-maximization [2]. Applying Algorithm 1, we derive all other SCMs within the equivalence class of $\mathfrak{C}$. Unique identifiability occurs only when all observed variables are isolated (see Remark 4). While we did not develop practical algorithms for steps [1] and [2] in this paper, they represent promising directions for future research.

---

> > ### Comment · Reviewer_AfH3 · 2024-12-02
> >
> > Thank you for your response. I will keep my original score.

---

### Official Review · Reviewer_tdTc · 2024-11-09

**Soundness:** 4
**Presentation:** 4
**Contribution:** 4
**Rating:** 8
**Confidence:** 3

**Summary:**

The paper establishes general conditions under which linear structured causal models are identifiable in the presence of hidden confounding variables and Gaussian noise. Briefly, if the number of confounding random variables is at most the number of observable random variables, then a finite number of equivalent models can be recovered from observations of the observables (the number of models is $n!$, where $n$ is the number of observables). If, in addition, the confounders are non-Gaussian, then the model is uniquely identifiable. Some minor technical assumptions are needed.

**Strengths:**

This is a very general result on a fundamental model of causality.

**Weaknesses:**

The result is information-theoretic. It assumes perfect statistics and does not analyze the resilience to perturbations due to finite sampling. Also, the paper does not state clearly any algorithmic consequences (complexity etc.), though the methods are mostly linear-algebraic and hence possibly effective.

**Questions:**

None

---

> ### Author Response · Authors · 2024-11-25
>
> Thank you for your constructive feedback.
>
> > Also, the paper does not state clearly any algorithmic consequences (complexity etc.), though the methods are mostly linear-algebraic and hence possibly effective.
>
> We added the following to determine the complexity of Algorithm 1:
>
> The computational complexity [of Algorithm 1] is dominated by operations on $n \times n$ matrices. The most intensive steps---matrix multiplication ($\Sigma = QQ^\top$), Cholesky decomposition of $\Sigma_p$, and inversion of $Q'$---each require $O(n^3)$ time. Thus, the overall complexity is $O(n^3)$.

---

### Meta-Review · Area_Chair_Jn6g · 2024-12-11

**Metareview:**

There is a clear consensus that this work provides strong results on Structural Causal Models and their identifiability conditions.  Specifically, suitable sufficient conditions are given for unique identifiability or finite identifiability, depending on the (non-)Gaussianity of the confounders, the number of them, and whether the causal structure is known.  The reviewers commented positively on essentially all aspects, including the results writing and clarity, and there was plenty of interaction between the authors and reviewers that culminated in the reviewers having no significant concerns in their final recommendation.

The reviewers particularly commented favorably on (i) the fundamental nature of the problem, (ii) the level of detail, rigor, and completeness in the results, (iii) the ease and enjoyment of reading the paper, and (iv) the quality of certain specific results such as Theorem 2.

**Additional Comments On Reviewer Discussion:**

Some questions/clarifications were answered by the authors and the reviewers were satisfied with the responses.  No reviewer discussion was needed.

---

### Decision · Program_Chairs · 2025-01-22

Accept (Spotlight)